# Self-repairing interphase reconstructed in each cycle for highly reversible aqueous zinc batteries

Wenyao Zhang[1,2], Muyao Dong[3], Keren Jiang [1], Diling Yang[1], Xuehai Tan [1], Shengli Zhai[1], Renfei Feng [4], Ning Chen [4], Graham King [4], Hao Zhang [1], Hongbo Zeng [1], Hui Li [3], Markus Antonietti [5] & Zhi Li [1] ✉

Aqueous zinc (Zn) chemistry features intrinsic safety, but suffers from severe irreversibility, as exemplified by low Coulombic efficiency, sustained water consumption and dendrite growth, which hampers practical applications of rechargeable Zn batteries. Herein, we report a highly reversible aqueous Zn battery in which the graphitic carbon nitride quantum dots additive serves as fast colloid ion carriers and assists the construction of a dynamic & self-repairing protective interphase. This real-time assembled interphase enables an ion-sieving effect and is found actively regenerate in each battery cycle, in effect endowing the system with single $Zn^{2+}$ conduction and constant conformal integrality, executing timely adaption of Zn deposition, thus retaining sustainable long-term protective effect. In consequence, dendrite-free Zn plating/stripping at ~99.6% Coulombic efficiency for 200 cycles, steady charge-discharge for 1200 h, and impressive cyclability (61.2% retention for 500 cycles in a Zn||$MnO_2$ full battery, 73.2% retention for 500 cycles in a Zn||$V_2O_5$ full battery and 93.5% retention for 3000 cycles in a Zn||$VOPO_4$ full battery) are achieved, which defines a general pathway to challenge Lithium in all low-cost, large-scale applications.

"Beyond lithium-ion" chemistries are eagerly demanded to develop safe, cost-effective, and reliable grid-scale energy storage technologies[1,2]. Among state-of-the-art electrochemical devices, rechargeable aqueous Zn-ion batteries are in principle promising because metallic Zn is globally available, environmentally benign, and insensitive in oxygen and humid atmosphere, and the two-electron redox reaction of $Zn/Zn^{2+}$ and low polarizability confer a high theoretical capacity (819 mAh g$^{-1}$ and 5855 mAh cm$^{-3}$) and power[3,4]. However, the long-standing roadblock of practical commercialization lies in the notorious side reactions of metallic Zn with water that are occurring at the electrode/electrolyte interface[5],

leading to a low Coulombic efficiency (CE), uncontrolled Zn dendrite growth, followed by quick short-circuiting.

Solvated $Zn^{2+}$ forms a stable hydration shell in water, which in turn constitutes a high energy barrier against desolvation. The acidic character inherently induces parasitic water reduction during Zn deposition process[6], whereas the generation of $H_2$ lets the pH value in local areas of the Zn electrode fluctuates. The increase in local pH corrodes the Zn surface and drives the formation of an insulating and passivating zincoxidic layer [7].

To weaken the reactivity of water and alleviate water-induced side reactions, a wide range of approaches was described[8–13]. Electrolyte

[1]Department of Chemical and Materials Engineering, University of Alberta, Edmonton T6G 1H9 AB, Canada. [2]Key Laboratory for Soft Chemistry and Functional Materials, Ministry of Education, Nanjing University of Science and Technology, 210094 Nanjing, China. [3]Beijing Advanced Innovation Center for Soft Matter Science and Engineering, Beijing University of Chemical Technology, 100029 Beijing, China. [4]Canadian Light Source, Saskatoon S7N 2V3 SK, Canada. [5]Colloid Chemistry Department Department, Max Planck Institute for Colloids and Interfaces, 14424 Potsdam, Germany. ✉e-mail: zhi.li@ualberta.ca

optimization has drawn intensively studied and the strategies mainly include "water-in-salt" electrolytes[14–17], deep-eutectic electrolytes[18,19], molecular-crowding electrolytes[13,20], and functional electrolyte additives[21–24]. Incorporating electrolyte additives is in the present context of particular interest. The role of most additives is to help to construct protective interphases that sustain high $Zn^{2+}$ ion conduction but suppress water penetration and decomposition[24,25], similar to the solid electrolyte interphase (SEI) protection mechanisms known from lithium (Li) metal batteries. The reduction of $Zn/Zn^{2+}$ however occurs at a less negative potential (−0.76 V vs. standard hydrogen electrode(SHE))[20,26,27] compared to $Li/Li^+$ (−3.04 V vs. SHE), where common anions or solvents are difficult to decompose reductively, while water splitting takes place, admittedly weakening any potential Zn-SEI. Such interphases formed by corrosion rather than polymerization are not robust but prone to detach. Several additives were proposed to regulate the solvation structure of $Zn^{2+}$ in the electrolyte[22,24]. To our knowledge, this suppression is built on the gradual consumption of additives that occurs during Zn deposition and is thereby not sustainable and falls short in long-term cycling. Lu and Archer et al.[28] reported in situ constructed interphases by involving graphitic carbon nitride nanosheets in colloidal electrolytes which achieve ordered assembly of metal electrodeposits. Zn-SEI formed by graphitic carbon nitride nanostructures can effectively promote spatially compact Zn deposits with high levels of reversibility, but the gradual weakened protective effect due to inevitable defects coming on Zn-SEI and the consumption of additives remain unsolved.

Here, we propose and demonstrate a conceptually new strategy to dynamically guide homogeneous Zn deposition and fundamentally eradicate Zn dendrite growth. The employed graphitic carbon nitride quantum dots ($C_3N_4$QDs) as an electrolyte additive are nanosheet-like and possess numerous periodic coplanar zincophilic pores intrinsically[29,30], which serve as fast colloidal carriers that endow high $Zn^{2+}$ conductivity and transference number, in effect moderating a more even distribution of $Zn^{2+}$ ion flux. Correspondingly, the $Zn^{2+}$ solvation structure is optimized via interacting with $C_3N_4$QDs, minimizing the inhomogeneity of Zn nucleation during initial plating. The in situ constructed interphase layer made up of the deposited $C_3N_4$QDs nanotiles separates the forming Zn metal from the reactive water, while keeping the pores for ion-sieving open to enable water-free, single $Zn^{2+}$ ion conduction. Notably, these interfacial $C_3N_4$QDs are bound by Coulombic forces to the metal surface but redisperse into the electrolyte upon potential inversion, showing dynamic regeneration in each battery cycle. In that, the $C_3N_4$QDs component is not consumed, which sustainably guarantees conformal integrality of the interphase. As proof of concept, the metallic Zn in aqueous $ZnSO_4$ electrolyte with $C_3N_4$QDs delivers an impressive Zn plating/stripping CE to 99.7% (over 200 cycles at 2 mA cm$^{-2}$, 1 mAh cm$^{-2}$) and long-term cycling stability of up to 1200 h (at 1 mA cm$^{-2}$, 1 mAh cm$^{-2}$). The dynamic carbon nitride SEI deposition technology thereby brings unprecedented reversibility to aqueous Zn batteries, finally demonstrated with either $V_2O_5$, $MnO_2$ or $VOPO_4$ cathodes.

## Results

### Fast ion carrier in electrolyte: strong interaction between $Zn^{2+}$ and subnanometric pores in $C_3N_4$

The functional electrolyte is achieved by dispersing a certain amount of $C_3N_4$QDs in 2 M $ZnSO_4$ aqueous electrolyte. A typical $C_3N_4$QD structure was presented in Supplementary Fig. 1, which is composed of the condensed tri-s-triazine (tri-ring of $C_6N_7$) subunits connected through planar tertiary amino groups, possessing periodic pores of ~0.68 nm in the lattice. As characterized by TEM and AFM, the as-synthesized $C_3N_4$QD have a typical nanoplate morphology with an average lateral size of ~10 nm and a thickness of 1.5 nm (Supplementary Figs. 1 and 2). It is known that the $C_3N_4$QDs are colloidally well dispersed as a result of charging oxygen and nitrogen groups on the

edge[31] (Supplementary Fig. 3). Our study reveals that the $C_3N_4$QDs remain excellent dispersibility and stability in $ZnSO_4$ aqueous electrolyte, no observable precipitation can be detected, at least over 8 months when the $C_3N_4$QDs content is below 1 mg mL$^{-1}$ (Supplementary Fig. 4). This might originate from the recharging of $C_3N_4$QDs by adsorbed Zn ions, screening interactions, and providing such high dispersibility even in high salt aqueous solutions.

The strong interaction between $Zn^{2+}$ and periodic subnanometric pores in $C_3N_4$ can be explored using Raman spectroscopy. As shown in Fig. 1a, the spectrum of pristine $ZnSO_4$ electrolyte exhibits a distinct shoulder peak at 385 cm$^{-1}$ assigned to the symmetrical stretching mode of the octahedral $[Zn(OH_2)_6]^{2+}$. This result underlines that $Zn^{2+}$ exists as a hexahydrate solvate under these conditions[32,33]. Upon introducing $C_3N_4$QDs, the $[Zn(H_2O)_6]^{2+}$ peak decreases and becomes broad. Given the symmetric (413 cm$^{-1}$) and asymmetric (418 cm$^{-1}$) stretching mode of Zn-N[34], this is attributed to a gradually weakened interaction of $Zn^{2+}$ with water and binding to $C_3N_4$QDs, proving the new coordination configuration for $Zn^{2+}$. Additionally, we detected an evident blueshift of the vibration stretching of $SO_4^{2-}$ after the addition of $C_3N_4$QDs on FT-IR spectra (Supplementary Fig. 5), which unveils lower binding of $SO_4^{2-}$ and thereby confirms further separation from the $Zn^{2+}$ coordination sheath. The change in $Zn^{2+}$ solvation configuration can be also evidenced by X-ray absorption near-edge structure (XANES) analysis. Zn K-edges of a sequence of $ZnSO_4$-$C_3N_4$QDs in Fig. 1b all shift to lower energy with reference to the pristine $ZnSO_4$ electrolyte, suggesting the electron transfer from Zn to O in $H_2O$ is efficiently restrained through introducing $C_3N_4$QDs. As further revealed by the $k^3$-weighted extended X-ray absorption fine structure (EXAFS) in Fig. 1c, the dominant peak at 1.66 Å from Zn-O slightly redshifts with increasing $C_3N_4$QDs content, verified again the weakened interaction between $Zn^{2+}$ and $H_2O$ and a reduced O-coordination around $Zn^{2+}$.

The coordinating number of $Zn^{2+}$ in $ZnSO_4$-$C_3N_4$QDs system was then modeled with molecular dynamics (MD) simulation (Fig. 1d). A periodic unit of $C_3N_4$QD with one subnanometric pore surrounded by three tri-s-triazine was chosen to constitute the cubic box of the electrolyte. Evident by the movement track of the $Zn^{2+}$ ion in the $C_3N_4$QDs-$ZnSO_4$ electrolyte in Supplementary Fig. 6, the intrinsic subnanometric pore in $C_3N_4$QD is identified as the most stable binding site for the $Zn^{2+}$. Typical solvation structure of $Zn^{2+}$ in $ZnSO_4$ is constituted by a $Zn^{2+}$ and six water molecules via stable octahedral coordination, whereas the $C_3N_4$QD can enter the primary solvation shell of $Zn^{2+}$ and replace two water molecules forming a $[Zn(C_3N_4)(H_2O)_4]^{2+}$ complex. The intrinsic subnanometric pore in $C_3N_4$QD serves as the most stable binding site for the $Zn^{2+}$. Additional proof is provided by the radial distribution functions (RDFs) (Fig. 1e), which confirm the coexistence of $Zn^{2+}$-O and $Zn^{2+}$-N coordination in $ZnSO_4$-$C_3N_4$QDs system, with bond lengths of 2.05 Å and 3.35 Å, respectively. The $Zn^{2+}$-O pair in $ZnSO_4$-$C_3N_4$QDs is slightly enlarged compared with that in $ZnSO_4$ (~1.94 Å)[22,35], in accordance with the above XANES results. The weakened bonding strength between $Zn^{2+}$ and $H_2O$ reduces the proton activity and suppresses electrochemical water decomposition, as also evidenced by a gradually lower hydrogen evolution potential with increasing doses of $C_3N_4$QDs in the electrolyte (Supplementary Fig. 7).

Density functional theory (DFT) calculations were conducted to unveil the interaction among $Zn^{2+}$ ion, $H_2O$, and $C_3N_4$QDs (Fig. 1f). From binding energy results, $[Zn(C_3N_4)(H_2O)_4]^{2+}$ possesses the optimal energy in the explored $Zn^{2+}$ solvation configurations. The broadened Zn orbitals and their overlaps with carbon and nitrogen orbitals in the density of states (DOS) analysis (Supplementary Fig. 8) explain the origin of strong interaction between $Zn^{2+}$ and $C_3N_4$QDs. The $[Zn(C_3N_4)(H_2O)_4]^{2+}$ also possess lower electrostatic potential (less charge transfer) along with a more uniform surface charge distribution (Fig. 1g). This finding allows us to deduce an enhancement on the $Zn^{2+}$ migration and transport, which is later quantitatively verified by the characterization for transference number of $Zn^{2+}$($t_{Zn2+}$, Supplementary

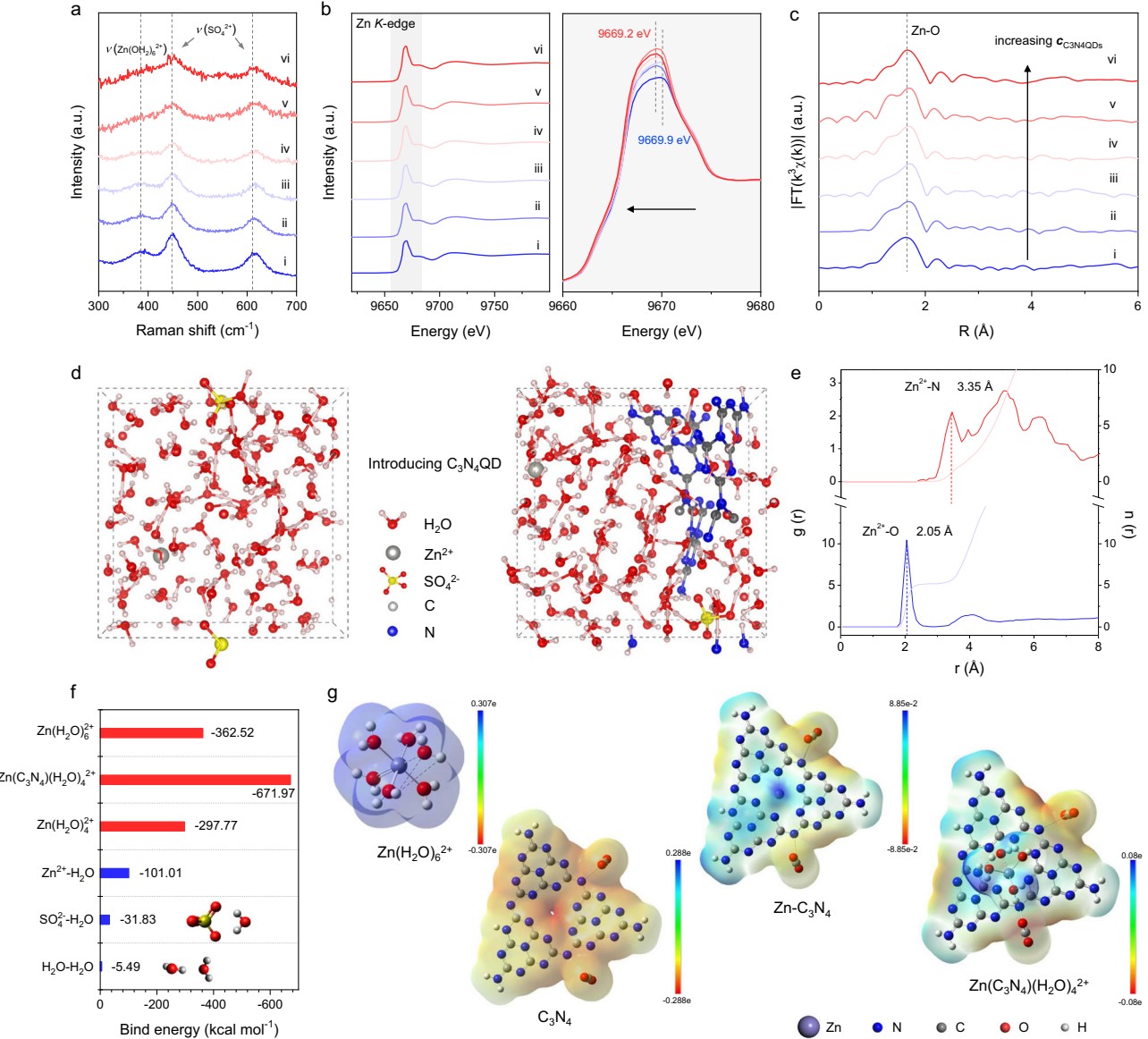

**Fig. 1 | The structural characterization of ZnSO$_4$-C$_3$N$_4$QDs electrolyte. a** Raman spectra, **b** XANES spectra and **c** Fourier transformed EXAFS spectra of (i) pristine 2 M ZnSO$_4$, (ii) 2 M ZnSO$_4$ + 0.1 mg ml$^{-1}$ C$_3$N$_4$QDs, (iii) 2 M ZnSO$_4$ + 0.5 mg ml$^{-1}$ C$_3$N$_4$QDs, (iv) 2 M ZnSO$_4$ + 1 mg ml$^{-1}$ C$_3$N$_4$QDs, (v) 2 M ZnSO$_4$ + 2 mg ml$^{-1}$ C$_3$N$_4$QDs, (vi) 2 M ZnSO$_4$ + 4 mg ml$^{-1}$ C$_3$N$_4$QDs; **d** The 3D snapshot of the ZnSO$_4$-C$_3$N$_4$QDs electrolyte obtained from MD simulations; **e** RDFs for Zn$^{2+}$-N (C$_3$N$_4$QDs) and Zn$^{2+}$-O (H$_2$O) in the ZnSO$_4$-C$_3$N$_4$QDs electrolyte from MD simulations; **f** the binding energies of Zn$^{2+}$ solvation configurations based on the DFT calculations; **g** the electrostatic potential distributions of [Zn(H$_2$O)$_6$]$^{2+}$, original C$_3$N$_4$QDs, Zn$^{2+}$-C$_3$N$_4$QDs, and [Zn(C$_3$N$_4$)(H$_2$O)$_4$]$^{2+}$ solvation structures.

Fig. 9). The ZnSO$_4$ electrolyte renders a low $t_{Zn2+}$ of 0.58, which imposes a nonnegligible Zn$^{2+}$ ion concentration gradient at the vicinity of Zn electrode and induces the buildup of a strong interface polarization[36,37]. On the contrary, $t_{Zn2+}$ in ZnSO$_4$-C$_3$N$_4$QDs, as listed in Supplementary Table 1, is substantially enhanced by 10–40%, depending on the C$_3$N$_4$QDs content. Besides surface polarization, we also have to consider the peculiar colloidal character of [Zn(C$_3$N$_4$)(H$_2$O)$_4$]$^{2+}$. As each intrinsic pore in C$_3$N$_4$QD can participate in metal coordination[38,39], a 10 nm C$_3$N$_4$QD is able to carry up to 185 Zn$^{2+}$ ions at the same time, thus a large amount of Zn$^{2+}$ undergoes synchronous migration under the electrical field. Such a transport behavior ensures a generous and homogeneous Zn$^{2+}$ ion flux, particularly at high current densities.

### Ion-sieving on Zn surface: horizontal placement of C$_3$N$_4$QDs with directional ion-channels

A consecutive question is how C$_3$N$_4$QDs interact and protect the metallic Zn anode. As presented in Supplementary Fig. 10, the C$_3$N$_4$QDs is calculated to adsorb on the Zn (002) crystal plane with high adsorption energy ($E_{ads}$ = −3.47 eV, horizontally), much higher than an H$_2$O molecule ($E_{ads}$ = −0.62 eV). This result suggests that the Zn electrode is coated with C$_3$N$_4$QDs rather than hydrated with water, which effectively inhibits reaction with water and the related corrosion. In addition, the rather stiff carbon nitride, as quantified by our previous studies[40], suppresses the generation of Zn protuberant tips and promotes homogeneity of Zn nucleation.

The interaction of [Zn(H$_2$O)$_6$]$^{2+}$ and [Zn(C$_3$N$_4$)(H$_2$O)$_4$]$^{2+}$ with Zn electrode is further modeled by moving diverse solvated Zn$^{2+}$ species along the Zn (002) surface (Supplementary Fig. 11). The [Zn(C$_3$N$_4$)(H$_2$O)$_4$]$^{2+}$ disclose larger potential energy and a smooth energy decrease curve in contrast to [Zn(H$_2$O)$_6$]$^{2+}$, suggesting faster Zn$^{2+}$ adsorption on the Zn surface.

The horizontal, coplanar placement of C$_3$N$_4$QDs is directly monitored by employing electrochemical atomic force microscopy (EC-AFM, Supplementary Fig. 12). The right image in Fig. 2a shows the

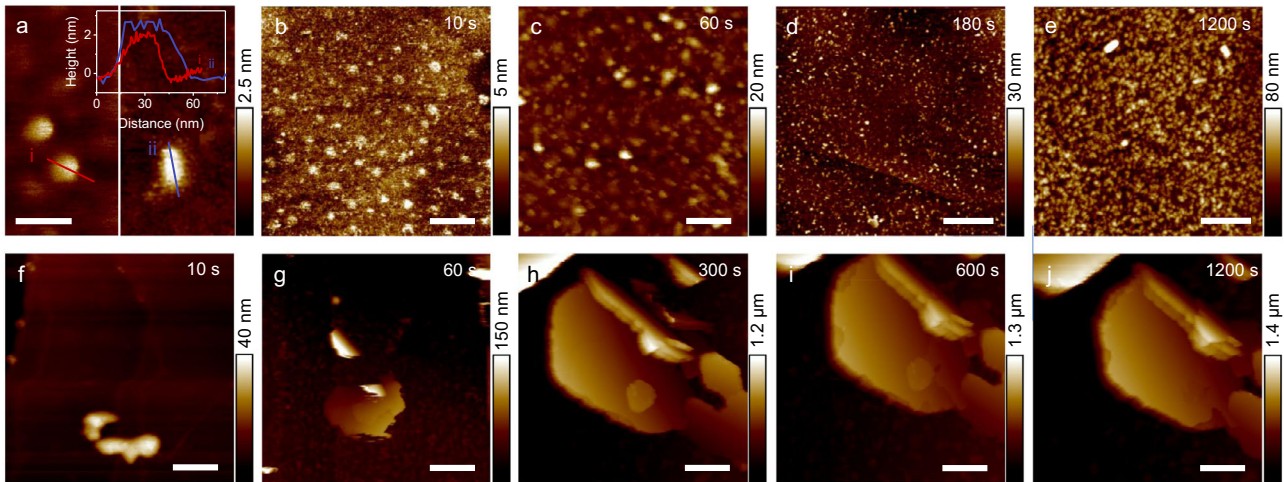

**Fig. 2 | The electrochemical behavior of $Zn^{2+}$ ions. a** Pristine $C_3N_4$QDs on mica (left) and the $C_3N_4$QDs in $ZnSO_4$ aqueous electrolyte on HOPG (right), the inset is the height profiles of the corresponding lines; In situ AFM images of Zn electrodeposits on HOPG with a current density of $100\,\mu A\,cm^{-2}$ in **b–e** 2 M $ZnSO_4$ + 0.5 mg mL$^{-1}$ $C_3N_4$QDs electrolyte and **f–j** 2 M $ZnSO_4$ electrolyte. Scale bar: 50 nm for **a**; 200 nm for **b**, **c**, **f**; 1 μm for **d**, **e**, **g–j**.

surface of a highly oriented pyrolytic graphite (HOPG) in EC-AFM cell with $ZnSO_4$-$C_3N_4$QDs electrolyte. Without even applying a voltage, we observed the coplanar adsorption of a $C_3N_4$QD with ~2 nm in height and ~20 nm in diameter, very similar to the pristine $C_3N_4$QD supported on dry mica (left image). The measured diameter of $C_3N_4$QD is enlarged due to the AFM tip convolution effect. The horizontal placement of $C_3N_4$QD on the electrode is critical for the electrode protection, finally forming the SEI by overlaying carbon nitride "tiles" or "plasters". It ensures that the periodic subnanometric pores in $C_3N_4$QD are oriented vertical to the electrode surface, serving as directional ion-sieving channels that only allow the pass of bare $Zn^{2+}$ ions, without coordinated water[41]. That eventually avoids the contact between electrodes with the active coordinated water and thereby significantly mitigates the water-induced side reactions.

After applying a voltage, in situ visualizations of the Zn electrochemical plating behavior in $ZnSO_4$-$C_3N_4$QDs electrolyte reveals even nucleation of Zn at the initial depositing stage (Fig. 2b) afterward the construction of a continuously thin $C_3N_4$QDs interphase on the substrate (Fig. 2c). Notably, the Zn deposits show a dense structure with only nanoscale surface fluctuation (Fig. 2d, e). On the contrary, for the conventional $ZnSO_4$, a terribly patchy Zn nucleation is found, shown in Fig. 2f. Considering that protrusions on the electrode exhibit a stronger electrical field, more Zn are preferentially deposited around these Zn nuclei rather than on smooth regions (Fig. 2g)[42]. This behavior self-amplifies throughout prolonged plating, and accordingly, individual Zn deposits with half-hexagonal morphology are observed in the studied zone, showing sharp edges and an uncontrolled vertical height. It is noteworthy that the crystals tend to grow along the Zn (002) plane in $ZnSO_4$ (Fig. 2h, i). The pros and cons of preferred orientation growth have been discussed before[43–45]. In this case, the random out-of-substrate Zn (002) with sharp edges gain the risk of piercing the separator or even shortening the battery.

**Real-time reconstructed process: $C_3N_4$QDs assembly regenerated in each cycle**
The surface chemistry of Zn electrode is evaluated by X-ray photoelectron spectroscopy (XPS) analysis. The Zn anodes after 15th stripping/plating process in symmetric cells using $ZnSO_4$ and $ZnSO_4$-$C_3N_4$QDs electrolyte (labeled as Zn@$ZnSO_4$ and Zn@$ZnSO_4$-$C_3N_4$QDs, respectively) are collected and characterized. The Zn@$ZnSO_4$ displays a distinct sulfate signal (Supplementary Fig. 13), indicative of the formation of a $Zn_4SO_4(OH)_6\cdot xH_2O$ deposit insulated the $Zn^{2+}$ ion flux[23]. This finding illustrates the passivation of Zn electrode occurring in

conventional $ZnSO_4$ electrolytes. In the case of Zn@$ZnSO_4$-$C_3N_4$QDs, no sulfate component can be detected, providing strong evidence for the suppression of corrosion and the ion-sieving effect of the $C_3N_4$QDs interphase. Moreover, the high-resolution C1s and N1s XPS spectra on Zn@$ZnSO_4$-$C_3N_4$QDs show evident C=C (284.6 eV), C=N (285.6 eV), C-N (287.9 eV), C-N=C (398.6 eV), and N-$(C)_3$ (399.7 eV) components, which are all well in line with the XPS peaks of the original $C_3N_4$QDs, demonstrating that $C_3N_4$QDs interphase with good structural integrity is formed (Fig. 3a, b). Further $Ar^+$ sputtering XPS depth profiling in Fig. 3c, d unveil that the chemical composition of C 1s and N 1s components remain almost unchanged for 600 s sputtering ($0.1\,nm\,s^{-1}$), suggesting a tiled multilayer structure of the SEI and confirming the $C_3N_4$QDs are efficiently involved in the Zn electro-deposition process. The in situ constructed $C_3N_4$QDs interphase on Zn electrode is up to 100 nm thick and considered internally stacked sheets bound via van der Waals attractions. It should be noted that the nitrogen content in the stripping side of Zn@$ZnSO_4$-$C_3N_4$QDs is found strikingly less than that in plating, attributed to the dynamic redispersion process of the $C_3N_4$QDs interphase under the positive surface potential of the applied electrical field.

This promising dynamic process was further investigated through in situ fluorescence microscopy observations. Illustrated in Fig. 3e, the $ZnSO_4$-$C_3N_4$QDs electrolyte exhibits strong green emission under UV irradiation. Upon electroplating, a prominent accumulation of $C_3N_4$QDs on Zn surface is observed, and the resulting $C_3N_4$QDs interphase also shows in fluorescence the desired uniformity and continuity. Homogeneous and dendrite-free Zn deposition processes can be monitored, while the fluorescent SEI layer is preserved. Interestingly, the following stripping step witnesses the disassembling and redispersion of the constructed $C_3N_4$QDs interphase into the electrolyte. Neither a peeling off nor aggregation phenomena can be monitored, illustrating the excellent redispersion of the $C_3N_4$QDs under positive potential conditions. This regeneration process by principle avoids the consumption of additives during prolonged cycling and delivers a sustainable long-term protective effect (Fig. 3f). Considering the less concentration of $C_3N_4$QDs, we calculated the percentage of the Zn ions that are influenced by $C_3N_4$QDs theoretically. As listed in Supplementary Table 2, even in an ideal situation, there will be only ~0.063% of $Zn^{2+}$ ions coordinated with $C_3N_4$QDs in electrolyte with 0.5 mg mL$^{-1}$ $C_3N_4$QDs. This extremely small percentage suggests that the impact of $C_3N_4$QDs on the overall $Zn^{2+}$ solvation structure would not be significant without the formation of dynamic SEI upon Zn anode.

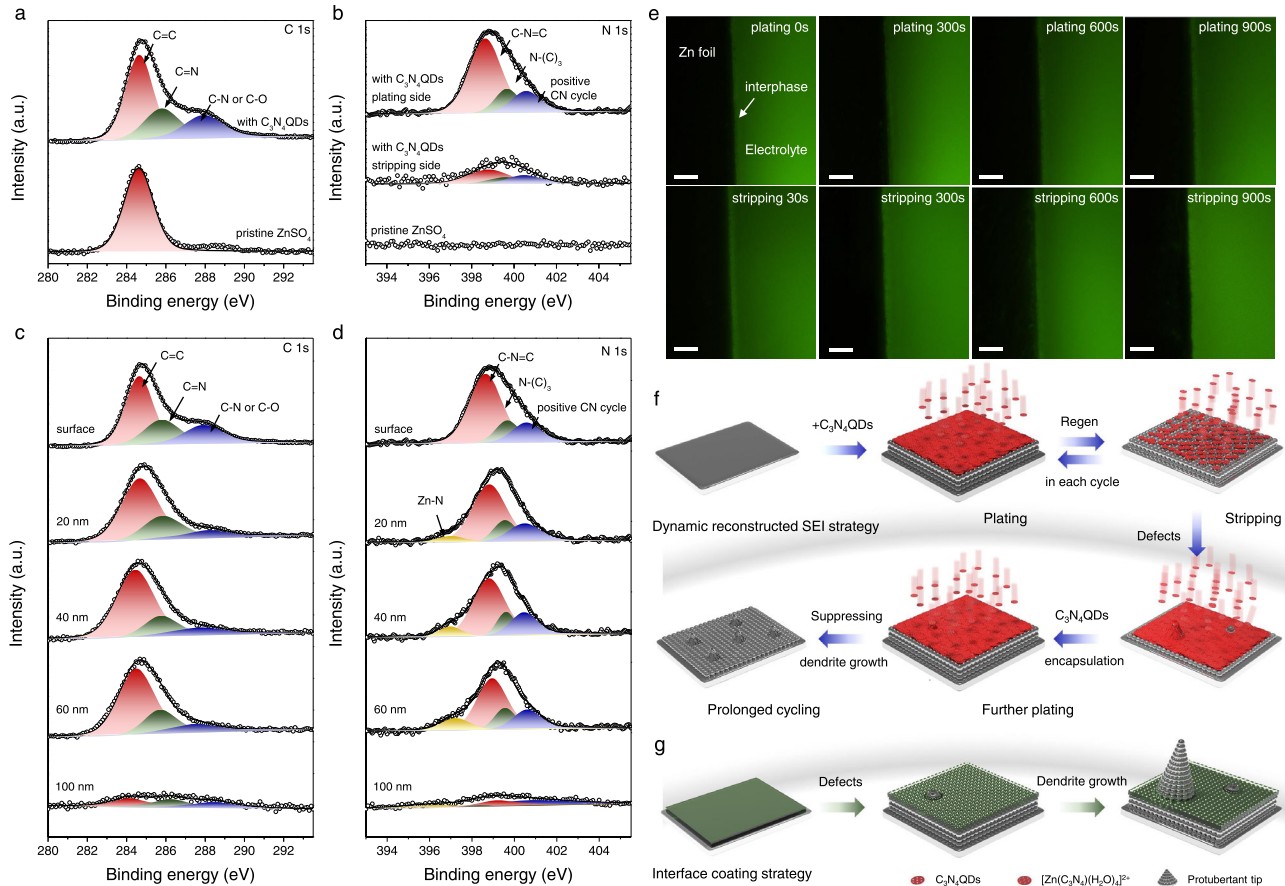

**Fig. 3 | The real-time reconstructed protective interphase regenerated in each cycle.** High-resolution XPS analyzation of the **a** C1s, and **b** N1s peaks obtained from the Zn anode after 15th stripping/plating process under the current density of 1 mA cm$^{-2}$ and capacity of 1 mAh cm$^{-2}$ in 2 M ZnSO$_4$ + 0.5 mg mL$^{-1}$ C$_3$N$_4$QDs electrolyte (top spectrum) and 2 M ZnSO$_4$ electrolyte (bottom spectrum); **c**, **d** the XPS depth profiles of the Zn anode after 15th stripping/plating process in 2 M ZnSO$_4$ + 0.5 mg mL$^{-1}$ C$_3$N$_4$QDs electrolyte; **e** in situ fluorescence microscopy images of Zn electrode during Zn plating/stripping process under 2 M ZnSO$_4$ + 0.5 mg mL$^{-1}$ C$_3$N$_4$QDs electrolyte, the scale bar is 50 μm; Schematic diagram of **f** real-time dynamic protective and **g** interface coating mechanism.

Furthermore, no matter how an SEI is created, there are always defective sites involved and trace protuberant tips generated at these sites. For the regular functional interphases, the defective sites remain at the same locations, and the tips gradually develop into large dendrites under cycling. Dynamic reconstructed interphase only allows the growth of protuberant tips in one cycle, since possible protuberant tips will be covered by the regenerated C$_3$N$_4$QDs in the following cycle and never grow into large dendrites. Such a real-time SEI reconstruction mechanism implies thereby self-adaption and self-healing, effectively maintaining constant conformal contact with Zn anode, timely correcting the Zn plating behavior, and intrinsically eradicating the irreversible fracture of the protective interphases, thereby guaranteeing a stable operation of Zn anode.

**Dendrite-free Zn deposition: C$_3$N$_4$QDs smoothen the deposition in various orientations and eliminate the byproducts**

The effects of ZnSO$_4$-C$_3$N$_4$QDs electrolyte on the texture control of metallic Zn deposits were analyzed by 2D synchrotron GIXD. Pristine Zn foil exhibits a GIXD pattern at 2θ angles of 36.52°, 39.16°, and 43.39°, assigned to the (002), (100), and (101) planes, respectively (Supplementary Fig. 14). The red-colored region between 5° and 20° is related to the elastic scattering[46]. After first stripping, the Zn@ZnSO$_4$-C$_3$N$_4$QDs present a similar GIXD pattern to pristine Zn, confirming its uniform Zn dissolution on each orientation (Fig. 4a). Conversely, the (101) plane in Zn@ZnSO$_4$ visibly exceeds, indicative of a preference of Zn dissolution in conventional ZnSO$_4$ (Fig. 4b). After first plating, there is an obvious scattering pattern located ~25° in Zn@ZnSO$_4$-C$_3$N$_4$QDs

corresponding to the inter-planar graphitic stacking of C$_3$N$_4$QDs, which strongly supports the deposition of the C$_3$N$_4$QDs protective interphase on Zn electrode (Fig. 4c). Isotropic Zn scattering with multiple discrete spots demonstrates a homogeneous plating process and a nanocrystalline feature of Zn deposits. On the contrary, Zn@ZnSO$_4$ discloses a strong (002) texture formation attributed to big, oriented crystal grains (Fig. 4d). The strengthened scattering stands for a preferred orientation growth of Zn, as also evidenced by in situ AFM, meanwhile, the enlarged spot testifies the structural distortion owing to the uneven plating. Nonnegligible GIXD patterns at 27.6°, 17.2° in Zn@ZnSO$_4$ index to the (006) and (004) plane of Zn$_4$SO$_4$(OH)$_6$ xH$_2$O byproduct, indicating a corrosion reaction occurs that triggers the passivation of Zn. Further conducting stripping/plating 15 cyclings showed the structural evolution from microsized Zn grain to nanocrystals on Zn@ZnSO$_4$-C$_3$N$_4$QDs, as noticed through the polycrystalline ring-like patterns in Fig. 4e. In the case of Zn@ZnSO$_4$, severe side reactions occur that give the electrochemically inactive Zn(OH)$_2$ and ZnO species (Fig. 4f). This finding confirms the crucial ion-sieving effect of the C$_3$N$_4$QDs protective interphase in circumventing the disadvantages of aqueous electrolytes, in consistent with the previous XPS analysis. When reaching the 75th cycle, Zn@ZnSO$_4$-C$_3$N$_4$QDs still undergo a uniform and straightforward Zn deposit, while Zn@ZnSO$_4$ turns worse and worse, reflecting self-amplification of irreversible processes (Fig. 4g, h). Notably, the surface of Zn electrode turns to be relatively rough and porous after the plating/stripping processes. Hence the X-rays' penetration depth enlarged and the intensity of Zn metal-based scattering signal

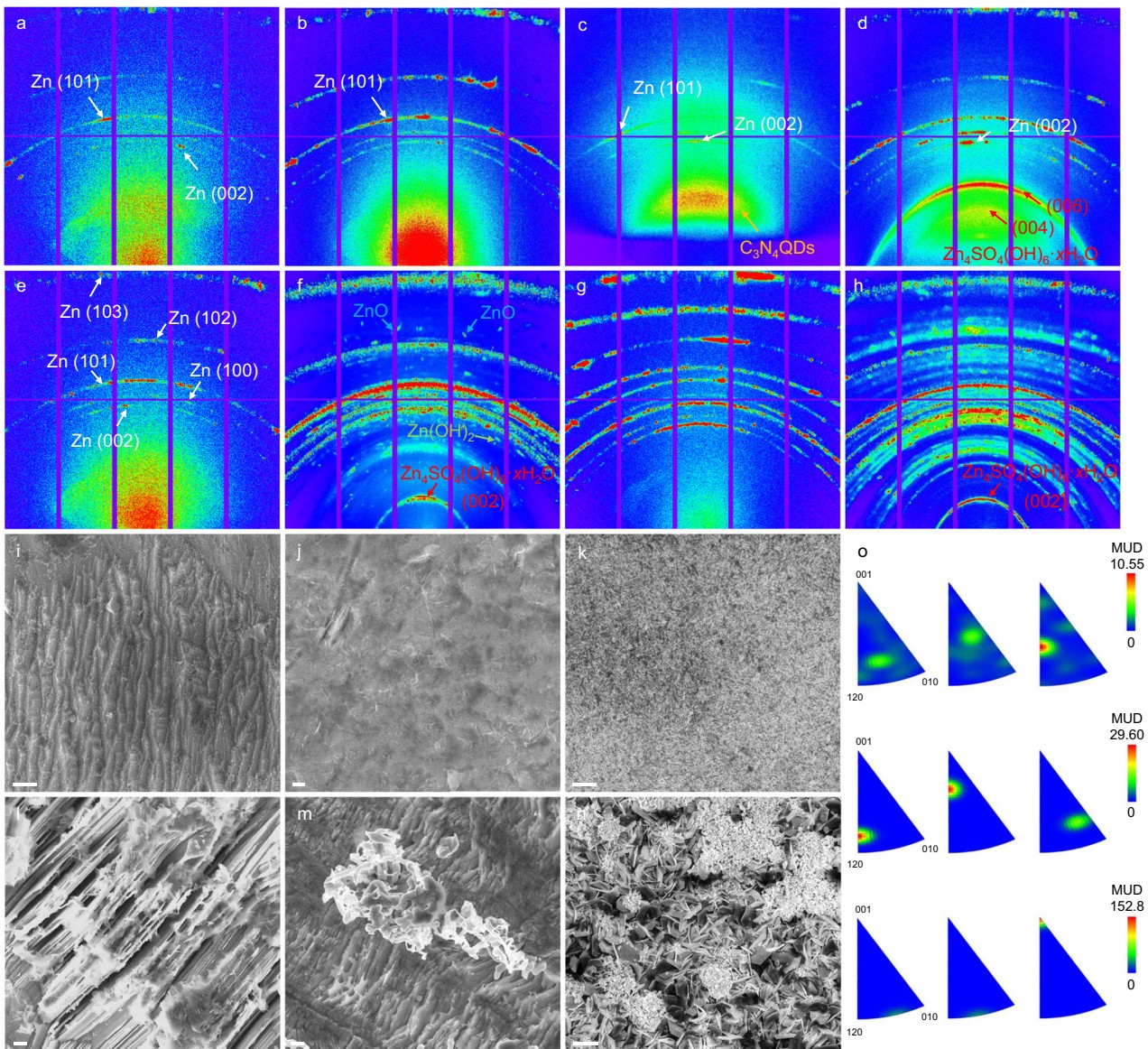

**Fig. 4 | Texture formation of Zn anodes.** 2D synchrotron grazing-incidence X-ray diffraction (GIXD) patterns of **a**, **c**, **e**, **g** Zn@ZnSO₄-C₃N₄QDs and **b**, **d**, **f**, **h** Zn@ZnSO₄ after 1st stripping, 1st, 15th, and 75th stripping/plating process under the current density of 1 mA cm⁻² with a capacity of 1 mAh cm⁻²; SEM morphologies of **i**–**k** Zn@ZnSO₄-C₃N₄QDs and **l**–**n** Zn@ZnSO₄ after 1st stripping, 1st, and 75th stripping/plating process; **o** inverse pole figures of the Zn deposits, Zn@ZnSO₄-C₃N₄QDs after 1st stripping/plating (top), Zn@ZnSO₄-C₃N₄QDs after 75th stripping/plating (middle) and Zn@ZnSO₄ after 75th stripping/plating process (bottom); Scale bar: 500 nm for **i**, **j**; 50 μm for **k**; 1 μm for **l**, **m**; 100 μm for **n**.

increased sharply. The scattering signal of the C₃N₄QD is not as evident as that obtained after 1st plating, however, it still can be recognized if compared with that of Zn@ZnSO₄ electrode.

Scanning electron microscopy (SEM) images further characterize the morphology evolution of Zn electrodeposits. As expected, the Zn@ZnSO₄-C₃N₄QDs unveils a ubiquitous coverage with anisotropic microstructures after first stripping, with the opposite to a loose distributed Zn (002) alignment on Zn@ZnSO₄ (Fig. 4i, l, Supplementary Figs. 15, and 16). Small-sized and dense-packed Zn is subsequently deposited on Zn@ZnSO₄-C₃N₄QDs instead of the typical out of substrate hexagonal Zn flakes (Fig. 4j, m and Supplementary Fig. 17), which implies the preferred orientations in the metallic structure of Zn can be tuned by the introduction of C₃N₄QDs. The jumbled stacking of Zn flakes on Zn@ZnSO₄ will feature aggressively extending dendrites and trigger the internal short circuit when reaching the Sand's time[47,48]. The observation after 75th stripping/plating has witnessed this hypothesis,

where the Zn@ZnSO₄-C₃N₄QDs maintains a smooth surface but the Zn@ZnSO₄ is full of cliffy dendrite pieces and dark by-product aggregations (Fig. 4k, n and Supplementary Figs. 18 and 19). Focused ion beam (FIB)-SEM analysis was then conducted to provide insights into the inner microstructural details of Zn deposition under different electrolytes. As displayed in Supplementary Fig. 20, hexagonal Zn platelets with various sizes are observed to form loosely connected building blocks assembled to create a porous electrodeposit structure on Zn@ZnSO₄. The Zn deposits are revealed to be randomly oriented. In contrast, the Zn@ZnSO₄-C₃N₄QD electrode exhibits dense Zn electrodeposits without inner interstices, no preferred oriented growth is detected and uniform nanoscale surface fluctuation is formed, in line with the in situ AFM observations. The concentration of C₃N₄QDs additive is also considered, a gradual improvement on reducing dendrites can be summarized, however, excess C₃N₄QDs accumulation on the interface disturbs the atomic diffusion,

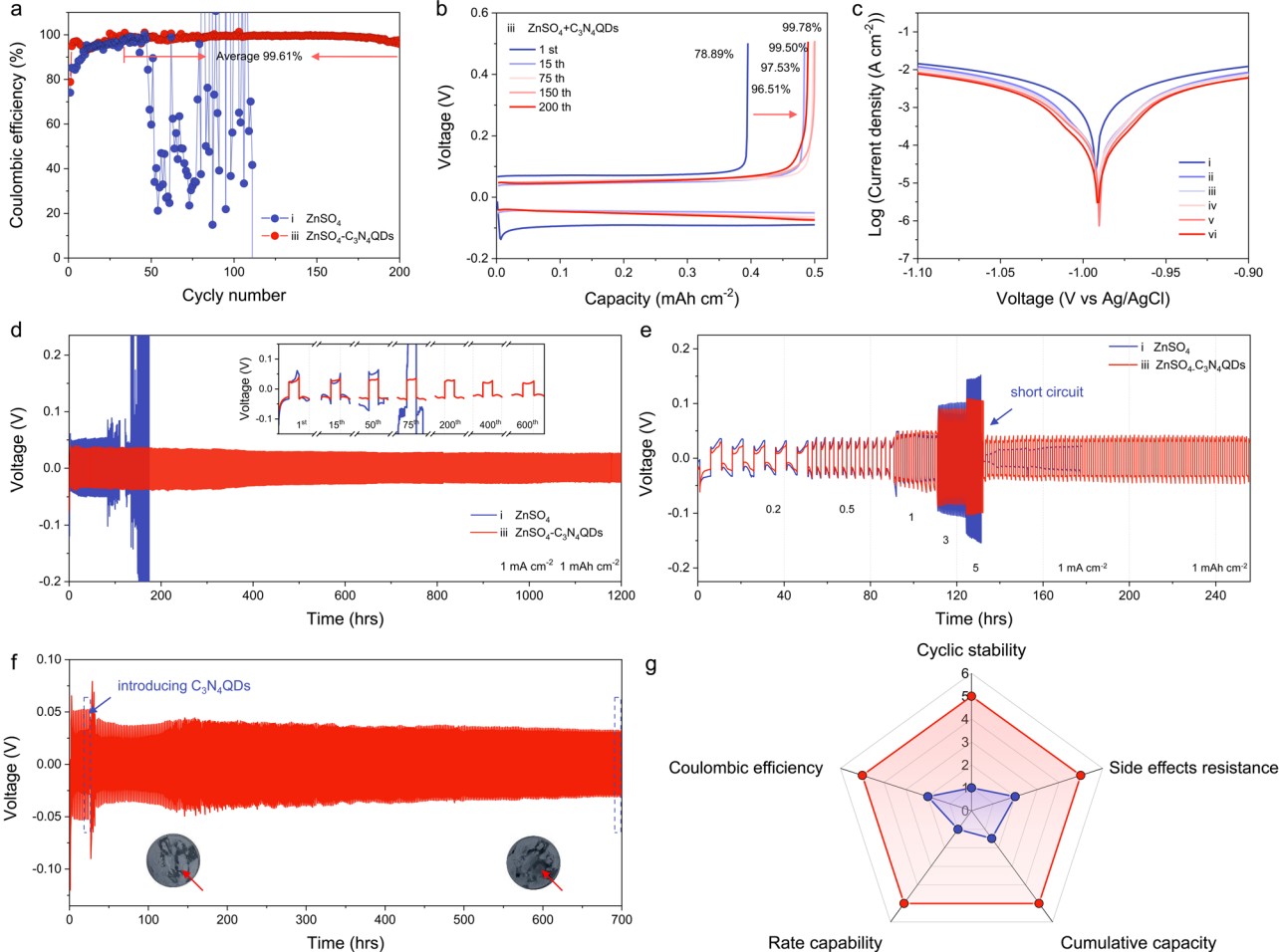

**Fig. 5 | Electrochemical Zn/Zn²⁺ reactions in ZnSO₄ and ZnSO₄-C₃N₄QDs electrolytes. a** Coulombic efficiency measurements of Zn||SS cells under different electrolytes system and corresponding voltage profiles obtained in **b** 2 M ZnSO₄ + 0.5 mg ml⁻¹ C₃N₄QDs electrolytes at various cycles; **c** linear polarization curves of Zn foil in different electrolytes; **d** comparison of long-term galvanostatic Zn stripping/plating in the Zn||Zn symmetric cells and time-voltage profiles under 1 mA cm⁻² with 1 mAh cm⁻²; **e** rate performance from 0.2 to 5 mA cm⁻²; **f** time–voltage profile in the dendrite involved Zn||Zn symmetric cell with introducing C₃N₄QDs under 1 mA cm⁻² with 1 mAh cm⁻²; **g** comparison of performance in terms of cyclic stability, coulombic efficiency, rate capability, cumulative capacity, and side effect resistance, The Roman numbers represent (i) pristine 2 M ZnSO₄, (ii) 2 M ZnSO₄ + 0.1 mg ml⁻¹ C₃N₄QDs, (iii) 2 M ZnSO₄ + 0.5 mg ml⁻¹ C₃N₄QDs, (iv) 2 M ZnSO₄ + 1 mg ml⁻¹ C₃N₄QDs, (v) 2 M ZnSO₄ + 2 mg ml⁻¹ C₃N₄QDs, (vi) 2 M ZnSO₄ + 4 mg ml⁻¹ C₃N₄QDs, respectively.

influencing the Zn²⁺ consumption and leading to crack formation (Supplementary Fig. 21). Additionally, the inverse pole figure in Fig. 4o clearly demonstrates the orientation change on the Zn deposits in the ZnSO₄-C₃N₄QDs electrolyte.

### Ultrastable Zn anode: effect of C₃N₄QDs in half-cell cycling

The reversibility of the proposed Zn electrochemistry in ZnSO₄-C₃N₄QDs electrolyte is evaluated using a Zn||Stainless Steel (SS) half-cell. Illustrated in Fig. 5a, the Zn plating/stripping behavior on SS undergoes a lattice fitting process in the first several cycles. After that, the Zn||SS half cell in ZnSO₄ delivers an average CE of 94.15% from 10 to 50 cycles, followed by drastically fluctuating signals owing to battery failure[18,49]. In contrast, Zn||SS half-cell in ZnSO₄-C₃N₄QDs shows high CE values from the initial stage and retains stability over 200 cycles. A high average CE of 99.61% is achieved, reflecting the close-to-full control of reactive water molecules along with the good reversibility of Zn in such electrolytes. It should be noticed that a slightly higher voltage polarization is detected when introducing C₃N₄QDs (Fig. 5b and Supplementary Fig. 22), indicating that dynamic SEI formation slightly lowers the ionic conductivity (Supplementary Fig. 23) and adds a resistive element (Supplementary Figs. 24–26). More additional electrochemical evidence is given by the linear polarization curves in

Fig. 5c, where positive-shift potentials and reduced currents are achieved in ZnSO₄-C₃N₄QDs, manifesting a low corrosion rate[50]. Electrochemical impedance spectroscopy (EIS) analysis elucidates the charge transfer resistance ($R_{ct}$) upon standing in ZnSO₄ progressively increases, which reflects the passivation of Zn anode, whereas it stays almost constant in ZnSO₄-C₃N₄QDs (Supplementary Figs. 27 and 28). The high interfacial stability also in mechanical terms provides homogeneous Zn²⁺ conduction pathways and accommodates volume change during the harsh electrochemical process. Importantly, the dynamic reconstructed C₃N₄QDs interphase ensures a regeneration process in each cycle, in sum resulting in a sustainable inhibiting effect on Zn dendrite growth.

The Zn||Zn symmetric cells under galvanostatic conditions were then probed for long-term cycling stability in ZnSO₄-C₃N₄QDs electrolytes. At 1 mA cm⁻² with 1 mAh cm⁻², the reference Zn@ZnSO₄ cell short-circuited within only ~160 h due to Zn dendrite growth (Fig. 5d). Upon integrating C₃N₄QDs, the cycle lifespans are all largely extended (Supplementary Fig. 29). Among various C₃N₄QDs concentrations, 0.5 mg ml⁻¹ is determined to be the optimal one, and all the studies in this work focus on this concentration. The inset in Fig. 5d depicts the voltage profiles of Zn@ZnSO₄-C₃N₄QDs, in which only a moderate voltage change is observed during the entire cycling, proving a mostly

reversible Zn plating/stripping process owing to the novel dynamic reconstructed process with $C_3N_4QDs$. The thickness of cycled cells is examined, and the presence of $C_3N_4QDs$ effectively reduces the inner pressure due to the parasitic water reduction (Supplementary Fig. 30). Even when cycled at high current densities of 3 mA cm$^{-2}$ and 5 mA cm$^{-2}$, the Zn@ZnSO$_4$-$C_3N_4QDs$ still demonstrate impressive cyclic stability over at least 1000 cycles (Supplementary Fig. 31). Analyzing the rate performance from 0.2 to 5 mA cm$^{-2}$ of Zn@ZnSO$_4$ (Fig. 5e) discloses erratic voltage responses with rapidly increasing hysteresis, testifying the continuous formation of detrimental byproducts. By contrast, the Zn@ZnSO$_4$-$C_3N_4QDs$ exhibits a steady rate capability, profiting from the good interfacial stability and favorable Zn$^{2+}$ conduction. When the current returns to 1 mA cm$^{-2}$ after 65 cycles, the voltage hysteresis on Zn@ZnSO$_4$-$C_3N_4QDs$ is recovered, suggesting high stability, which lays the foundation for practical Zn ion batteries.

The role of $C_3N_4QDs$ in suppressing the growth of dendrites is most impressively cross-tested by a designed self-repair experiment. The Zn anode is first cycled in only ZnSO$_4$ electrolyte for generating Zn dendrites (the inset in Fig. 5f). This dendritic Zn anode is then assembled into the ZnSO$_4$-$C_3N_4QDs$ electrolyte. The polarization voltage initially increases due to the instant accumulation of $C_3N_4QDs$ on the Zn dendrites at tip-strengthened electrical field points (Fig. 5f)[51]. After these initial phases, a steady voltage profile and a prolonged cycling life is retained. The dynamic reconstructed $C_3N_4QDs$ interphase is obviously able to repair the formerly ready-to-fail battery electrode by diverting uniform Zn$^{2+}$ deposition to adjacent regions until a smooth Zn deposition is reformed. To corroborate the improved overall performance raised by the $C_3N_4QDs$' self-repair&protection strategy, the comparisons of performance parameters is described in the radar chart (Fig. 5g). The Zn@ZnSO$_4$-$C_3N_4QDs$ deliver charming multifunctionality including outstanding cyclic stability, CE, resistance against side effects, rate performance along with high cumulative capacity, significantly outperforming in all aspects those of Zn@ZnSO$_4$.

## Long-lasting Zn-ion batteries: impact of $C_3N_4QDs$ in full-cell

Finally, we explored the application of the $C_3N_4QDs$ electrolyte additive in Zn-ion full batteries composed of a metallic Zn anode and a $V_2O_5$, $MnO_2$ or $VOPO_4$ cathode. Therein, conventional $V_2O_5$-based full cells were fabricated in ZnSO$_4$ electrolytes to demonstrate the role of $C_3N_4QDs$ (Supplementary Fig. 32). The initial discharge capacity of Zn| $C_3N_4QDs$|$V_2O_5$ cell is 153 mAh g$^{-1}$, which increases to the maximum of 170 mAh g$^{-1}$ at the 80th cycle and remains 112 mAh g$^{-1}$ after 500 cycles, 85 mAh g$^{-1}$ after 1000 cycles. In contrast, the Zn||$V_2O_5$ cell can only retain a capacity of 80 mAh g$^{-1}$ after 500 cycles, 52 mAh g$^{-1}$ after 1000 cycles. $MnO_2$ nanofibers are synthesized through a hydrothermal approach (Supplementary Fig. 33)[52]. Cyclic voltammetry (CV) profiles of Zn||$MnO_2$ full battery in aqueous ZnSO$_4$ + MnSO$_4$ electrolytes, with and without the presence of $C_3N_4QDs$, exhibits a similar Zn storage/ delivery response with distinct Mn-ion redox peaks (Supplementary Fig. 34)[53]. Relative to Zn||$MnO_2$, an enhanced current density is observed for Zn|$C_3N_4QDs$|$MnO_2$ alongside remarkable shifts of the cathodic/anodic peaks to more positive/negative voltages. It implies that well-dispersed $C_3N_4QDs$ provide accelerated reaction kinetics. This can be also deduced from the stepwise charge-discharge curves in Fig. 6a via a prevailing initial reversible capacity of 281.3 mAh g$^{-1}$ at 0.1 C and reduced voltage hysteresis. Further evidence is given by the EIS analysis with an $R_{ct}$ of ~79 Ohm for Zn|$C_3N_4QDs$|$MnO_2$, significantly lower than that of Zn||$MnO_2$ (~420 Ohm, Supplementary Fig. 35). Accordingly, the Zn|$C_3N_4QDs$|$MnO_2$ battery demonstrates a much better long-term stability with superior capacity retention after 500 cycles at 1 C (Fig. 6b). In sharp contrast, the Zn||$MnO_2$ battery undergoes fast capacity degradation. This probably is related to the irreversibility issues of metallic Zn anode all discussed above. The rate performance of both Zn|$C_3N_4QDs$|$MnO_2$ and Zn||$MnO_2$ illustrated drastically attenuated capacity with the increase of cycle number and

current density, which is caused by the serious decomposition/dissolution of $MnO_2$ cathode[54] (Supplementary Fig. 36). The Zn|$C_3N_4QDs$| $MnO_2$ exhibits slightly enhanced capacity retention.

We further investigate the applicability of the $C_3N_4QDs$ electrolyte additive with a recently developed vanadium-based cathode[55,56]. $VOPO_4$ lamellae (characterized in Supplementary Fig. 37) is employed to couple with the metallic Zn anode in 3 M aqueous Zn(OTf)$_2$ electrolyte under the addition of 0.5 mg ml$^{-1}$ $C_3N_4QDs$. Figure 6c compares the typical CV curves of Zn||$VOPO_4$ and Zn|$C_3N_4QDs$|$VOPO_4$ over a voltage range of 0.5–2 V. Three pairs of redox peaks at 1.15/1.46, 1.33/ 1.55, 1.53/1.68 V correspond to the multiple redox reactions of V$^{5+}$ to V$^{4+}$ in $VOPO_4$, and the ones at 1.84/1.90 V are assigned to the redox process between O$^{2-}$ and O$^{-}$[55]. $C_3N_4QDs$ also here improve the kinetics, as validated by higher response current densities in CV curves, combined with the effectively lowered $R_{ct}$ value in EIS analysis (Supplementary Fig. 38). As a result, the Zn|$C_3N_4QDs$|$VOPO_4$ witnesses miraculously stable operation over 3000 cycles at 1 A g$^{-1}$ with a capacity retention of 86.1% (Fig. 6d). Rate analysis of Zn|$C_3N_4QDs$|$VOPO_4$ presents good capacity retention with the increase of current as well as over cycling (Supplementary Fig. 36). After harsh 75 cycles, it can be noted that the Zn|$C_3N_4QDs$|$VOPO_4$ shows a recovered capacity of 115 mAh g$^{-1}$ when the current density is set back to 0.1 A g$^{-1}$. In contrast, the Zn||$VOPO_4$ gradually downgraded, which is in line with the cycling results in Fig. 6d. It is known that the decomposition/dissolution of $VOPO_4$ in aqueous electrolytes is usually the main issue leading to capacity/ voltage fading[57]. $C_3N_4QDs$ might also play a positive role in advancing these deficiencies of the $VOPO_4$ cathode, i.e. the dynamic SEI formation could work on both electrodes (Supplementary Fig. 39). Additional XPS analysis of the $VOPO_4$ cathode after the 15th cycle was performed (Supplementary Fig. 40), which clearly reveals an evident N 1s signal indicative of the formation of the $C_3N_4QDs$ interphase. This protective $C_3N_4QDs$ interphase also induces an ion-sieving effect to enable single Zn$^{2+}$ conduction, which can restrain the water activity and water-induced side reaction to a certain extent, significantly reducing the dissolution of $VOPO_4$. Most importantly, the protonophilicity of $C_3N_4$[58] favors capturing the protons thereby downgrading the protonation penetration depth. Combining the rather stiff nature of $C_3N_4$[40], the formation of $C_3N_4QDs$ interphase might maintain a rigid structure of $VOPO_4$ during insertion and deinsertion, hence preserving a stable cycling performance.

The highly stable and reversible structure of $VOPO_4$ in Zn(OTf)$_2$-$C_3N_4QDs$ was then studied by in situ Raman and ex-situ Wide-angle X-ray scattering (WAXS) techniques. As illustrated in Fig. 6e, the stretching vibration of P–O at 916.32 cm$^{-1}$, V=O at 993.8 cm$^{-1}$, and V–O–P at 1034.2 cm$^{-1}$ (labeled with triangles) alongside their bending modes (labeled with square)[59,60] gradually weaken during discharge, which ascribes to the Zn$^{2+}$ extraction, the distortion of VO$_6$ octahedra, and the reduction of V$^{5+}$. Impressively, the subsequent charging response presents the recovery of such Raman signals, energetically verifying a reversible Zn$^{2+}$ insertion/extraction process. During the entire discharging/charging, no new phases are detected through WAXS (Fig. 6f), indicating a typical Zn-ion storage mechanism in Zn| $C_3N_4QDs$|$VOPO_4$ system[55]. Upon fully discharged, the interlayer distance of $VOPO_4$ decreases owing to the strong electrostatic attraction between inserted Zn$^{2+}$ and oxygen in $VOPO_4$, showing a negative shift of the (001) scattering pattern in Fig. 6f and Supplementary Fig. 41. Of particular note is it increases upon charging, giving a deep understanding of the crystal-structure evolution of $VOPO_4$. In addition, the open circuit–voltage decay of fully charged cells followed by discharging after 48 h rest is monitored to probe the effect of $C_3N_4QDs$ on parasitic reactions in Zn||$VOPO_4$ full batteries (Supplementary Fig. 42). 93.52% of the original capacity can be retained in Zn|$C_3N_4QDs$|$VOPO_4$, far outperforming 67.03% in Zn||$VOPO_4$, which impressively illustrates the inhibition of the parasitic H$_2$ and O$_2$ evolutions after engaging the $C_3N_4QDs$ additive.

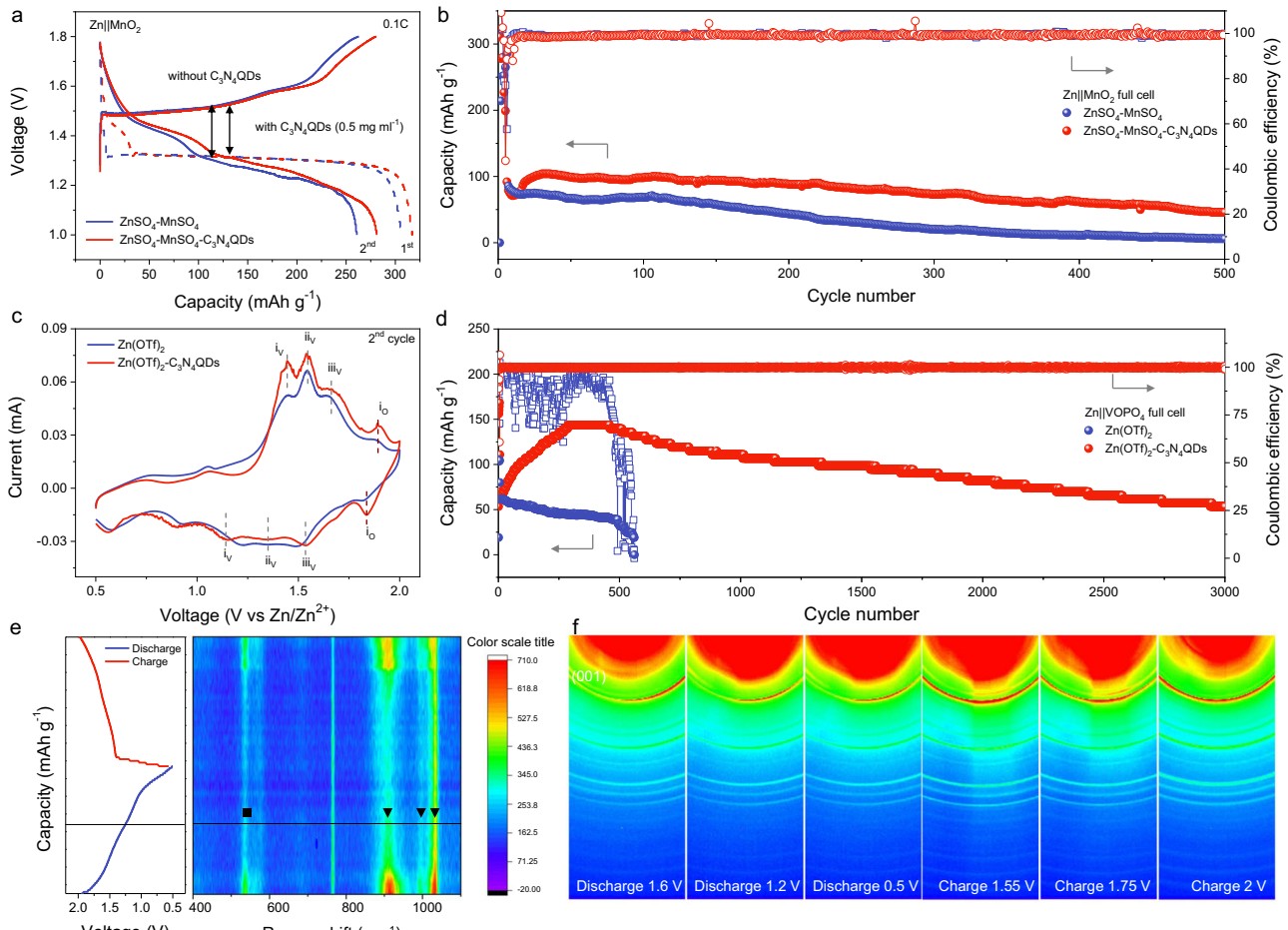

**Fig. 6 | Electrochemical performance of Zn-ion full batteries. a** Typical voltage profiles of Zn||MnO₂ and Zn|C₃N₄QDs|MnO₂ batteries at 0.1 C; **b** cyclic performance of Zn||MnO₂ and Zn|C₃N₄QDs|MnO₂ batteries at 1 C; **c** CV curves of Zn||VOPO₄ and Zn|C₃N₄QDs|VOPO₄ batteries at a scan rate of 0.1 mV s⁻¹; **d** cyclic performance of Zn||VOPO₄ and Zn|C₃N₄QDs|VOPO₄ batteries at 1 A g⁻¹; **e** in situ Raman spectra and the corresponding discharge/charge curves of Zn|C₃N₄QDs|VOPO₄, the triangle marks represent the stretching vibration of P–O at 916.32 cm⁻¹, V = O at 993.8 cm⁻¹, and V–O–P at 1034.2 cm⁻¹, respectively, and the square mark represents their bending modes; **f** two-dimensional WAXS patterns of VOPO₄ cathode at different voltage states in Zn(OTf)₂-C₃N₄QDs electrolyte.

## Discussion

In summary, we present an economic and efficient strategy to moderate aqueous Zn chemistry *via* introducing a C₃N₄QDs electrolyte additive. MD simulations combined with spectroscopic studies verified the strong interaction between C₃N₄QDs and Zn²⁺ ions, optimizing the Zn²⁺-solvation structure and decreasing the activity of solvated water molecules. C₃N₄QDs act as fast ion carriers to uniformize Zn²⁺ ion flux and harmonize ion migration. The in situ EC-AFM visualizations and GIXD analysis reveal an effective Zn²⁺ electrodeposition behavior in presence of C₃N₄QDs, essentially eradicating the uneven nucleation and preferred orientation growth. Fluorescence microscopy and XPS investigation unveil the in situ assemblies of layered C₃N₄QDs interphase on Zn anode during the plating process from its colloidal building blocks, while maintaining the ion-sieving effect of the single sheets to assure single Zn²⁺ conduction. This effectively suppresses the electrolyte-related interfacial side reactions. Importantly, this protective C₃N₄QDs interphase spontaneously redisperses into the electrolyte when reversing the electrical field upon stripping. Such a dynamic reconstruction mechanism maintains the integrality of the C₃N₄QDs interphase, circumvents the consumption of C₃N₄QDs, and delivers a sustainable long-term protective effect. As a result, dendrite-free and intrinsically stable Zn plating/stripping can be realized in the electrolyte with C₃N₄QDs. The self-healing mechanism is so strong that it even allows partial self-repair of reference cells without C₃N₄QDs additive. Finally, the C₃N₄QDs additive was also applied to Zn-ion full batteries with V₂O₅, MnO₂ and VOPO₄ cathodes, which deliver high levels of specific capacity, CE, and laudable capacity retention after long-term cycling. We believe that the concept of a dynamic real-time reconstructed SEI built from functional colloidal building blocks can be extended to other multivalent ion batteries that are often plagued with poor reversibility and sluggish kinetics, providing a brand-new route for the development of advanced energy storage devices.

## Methods

### Preparation of electrolytes

The g-C₃N₄ QDs were synthesized through thermal polymerization according to the method described by Shao[61]. A mixture of 2.3 g thiourea and 2.1 g citric acid was placed into an oven and calcined at 200 °C for 30 min. The product was first repurified by a 0.22 μm Milli-Q filter membrane and subsequently an MWCO = 500 dialysis bag for 2 days. The C₃N₄QD-ZnSO₄ electrolyte was prepared by adding 0.1–4 mg mL⁻¹ C₃N₄QDs in a conventional 2 M ZnSO₄ electrolyte with stirring for 1 h.

### Preparation of V₂O₅, α-MnO₂ and VOPO₄ cathode

V₂O₅ was purchased from Sigma-Aldrich and used without further purification. α-MnO₂ was fabricated by a hydrothermal method[52].

Typically, 1.5 mM $MnSO_4$ $H_2O$ and 1 mL 0.5 M $H_2SO_4$ were added into 45 mL of deionized water. After adding with 10 mL 0.1 M $KMnO_4$, the mixture was stirred for 2 h and then transferred to an 80 mL Teflon-lined autoclave at 120 °C for 12 h. The obtained $\alpha$-$MnO_2$ nanowires were collected by filtration, washed with water and isopropanol, and dried at 60 °C. The $\alpha$-$MnO_2$ cathode used here comprises 80 wt% $\alpha$-$MnO_2$, 10 wt% Super P carbon, and 10 wt% polyvinylidene fluoride (PVDF), which were mixed and well dispersed in N-methyl-2-pyrrolidone (NMP) and cast onto a 12 mm Stainsteel (SS) current collector. The active mass loading of the $\alpha$-$MnO_2$ cathode is ~1.5 mg cm$^{-2}$. $VOPO_4$ was fabricated by a reflux method[56]. In all, 4.8 g $V_2O_5$ 26.6 mL concentrated $H_3PO_4$ (85%) and 115.4 mL deionized water were mixed and refluxed at 115 °C for 20 h. The obtained $VOPO_4$ was collected by filtration, washed with water and acetone, and dried at 60 °C. The $VOPO_4$ cathode was prepared by the same procedure with 70 wt% $VOPO_4$, 20 wt% Super P carbon, and 10 wt% PVDF. The active mass loading of the $VOPO_4$ cathode is ~2 mg cm$^{-2}$.

## Electrochemical measurements

Zn foils with a diameter of 12 mm and a thickness of 200 μm as the electrode and a piece of glass fiber (GE-Whatman) as a separator were assembled into a CR-2032 type coin cell in an open environment. 120 μL of the corresponding electrolyte were added. Electrochemical cycling tests in Zn||Zn symmetric cells, Zn||SS cells, Zn||$\alpha$-$MnO_2$, Zn||$V_2O_5$, and Zn||$VOPO_4$ cells were recorded on a multichannel-current static system (Arbin Instruments BT 2000, College Station, TX, USA). The cyclic voltammetry (CV) and linear sweep voltammetry (LSV) were conducted on a VMP-300 electrochemical workstation (EC-lab, Biologic). The corrosion, diffusion, and hydrogen evolution behaviors of Zn foil anode were performed by a three-electrode system (Zn foil as working electrode, Pt as the counter electrode, and Ag/AgCl as reference electrode) on the VMP-300 electrochemical workstation. The corrosion Tafel plot was recorded by performing LSV with a potential range of ±0.3 V vs. open-circle potential of the system at a scan rate of 1 mV s$^{-1}$. The diffusion curves were measured by chronoamperometry method under an overpotential of −150 mV. The hydrogen evolution performance was collected through LSV with a potential range of −0.9−−1.6 V vs. Ag/AgCl in the 2 M $Na_2SO_4$ electrolyte with or without $C_3N_4$QD additives at a scan rate of 1 mV s$^{-1}$.

## Materials characterization

The morphologies of samples were characterized by field emission scanning electron microscopy (Zeiss Sigma 300), transmission electron microscopy (JEOL JEM-ARM200CF), and elemental mapping (Gatan GIF). Fourier transfer infrared (FT-IR) spectra were recorded on a Thermo Nicolet iS50 FTIR spectrometer ranging from 500 to 4000 cm$^{-1}$ under ATR mode. Raman spectra were conducted on a Renishaw InVIa Raman Microscope. X-ray photoelectron spectroscopy (XPS) measurements and depth profile XPS were performed on a Versa probe III (PHI 5000) spectrometer. Analysis was done using CASA XPS. All the XPS spectra were calibrated to the adventitious hydrocarbon (AdvHC) carbon peak at 284.8 eV. X-ray absorption spectroscopy was performed at Hard X-ray MicroAnalysis (HXMA, 06ID-1) beamline at Canadian Light Source. In situ EC-AFM was carried out on a Dimension Icon (Bruker) AFM and a coated conductive AFM probe (force constant of 5 N m$^{-1}$). A liquid model cell was used, in which the freshly cleaved highly oriented pyrolytic graphite (HOPG) (ZYH type, Bruker Corp.) was used as the working electrode (0.5 cm$^2$ exposed to the electrolyte), and the Zn strip was used as the counter and reference electrodes. In situ AFM images were obtained at different plating times at a constant current density of 0.2 mA cm$^{-2}$. All AFM images were acquired by contact mode scanning and each image took 260 s. The 2D synchrotron grazing-incidence X-ray diffraction (GIXD) was collected at Very Sensitive Elemental and Structural Probe Employing Radiation Beamlines (VESPERS, 07B2-1) at Canadian Light Source. The advantage of

GIXD technique on flat Zn electrodes is the limited penetration depth of the X-rays into the samples, with the benefit of low background scattering from the substrate. By varying the incident angle, the X-rays' penetration depth can be changed from a few nanometers up to 100 nanometers. The energy of X-ray beam used for GIXD is 8 keV and the beam size is ~5 × 7 μm. The GIXD patterns were measured in the incident angle range of 0.2°−2.6°, and the ones measured at 0.8° were shown above. The beam diffracted from the Zn foil was collected by a 2D area detector centered at 40° and located 120 mm away from the sample, which covers a $2\theta$ angular range of 6−73°. The powder diffraction data was collected at the Brockhouse High Energy Wiggler beamline of the Canadian Light Source using 35 keV X-rays. The 2D images were collected using a Perkin Elmer area detector. After cycling the batteries, the samples were taken out from the coin cell, carefully washed, and loaded into Kapton capillaries with an inner diameter of 1.2 mm.

## Molecular dynamics simulation

The MD simulation of $ZnSO_4$-$C_3N_4$QDs was performed in a cubic box (15.47 × 15.47 × 15.47 Å$^3$) of electrolyte with the periodic boundary condition, containing 100 $H_2O$, 1 $ZnSO_4$, and 1 $C_3N_4$QD, respectively. The basic model of the $C_3N_4$QDs is established according to Lu's work[61], and the structure of $C_3N_4$QDs is presented in Supplementary Fig. 8. The PBE functional[62] combined with the Goedecker–Teter–Hutter (GTH) pseudopotentials[63] and DZVP-MOLOPT basis sets (with the energy cutoff at 280 Ry) was employed to describe the electronic structure. The Grimme dispersion correction[64] was also included to better describe the intermolecular Van der Waals interactions. The time step for the MD simulation was set as 1.0 fs. The simulation system was firstly equilibrated in an isothermal–isobaric (NPT) ensemble (300 K, 1 atm) for 2 ps, and then relaxed in a canonical (NVT) ensemble (300 K) for 30 ps. The MD simulations were performed with the CP2K package[65], and the snapshots and radial distribution functions (RDFs) were analyzed based on the VMD software[66]. The electrostatic potential distributions were calculated at B3LYP/6-31 G* level using the Gaussian 09 program[67].

# Data availability

All data are available within the Article and Supplementary Information or available from the first author and corresponding author on reasonable request.

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

## Acknowledgements

This work is funded by NSERC Alliance (ALLRP 571058-21) - Alberta Innovates Advance (212200888) program. The authors acknowledge the nanoFAB facility at the University of Alberta for their material characterization support, and the national research facility at the Canadian Light Source for the XAS characterization on HXMA beamline, GIXD characterization on VESPERS beamline, and WAXS characterization on BXDS beamline.

## Author contributions

Z.L. supervised the project; W.Z., M.A., and Z.L. proposed the concept and initiated the project; W.Z. designed and performed the experiments and the characterizations; M.D. and H.L. performed the calculations; W.Z. and K.J. performed the in-situ AFM characterization; D.Y., H. Zhang, and H. Zeng performed the Zeta potential analysis; R.F., N.C., and G.K. helped to design and perform the synchrotron measurements at CLS, and helped to analyze the data; X.T. and S.Z. helped to discuss and analyze the data; all authors discussed and revised the manuscript.

## Competing interests

The authors declare no competing interests.
