## [Peer Review File · Nature Communications]

Reviewer comments, first round review

Reviewer #1 (Remarks to the Author):

I read with interest the manuscript by Zhang et al. reporting on using a dynamic interphase formed by graphitic carbon nitride as an electrolyte additive to improve the performance of Zn metal anodes. The concept appears to be scientifically sound. The authors also used complementary characterization tools (those at beamlines) to investigate the process. Overall, this work appears to be of interest to the readership of Nature Comm, and I thus suggest that authors further revise the manuscript to address the issues as detailed below.

1. The authors claim the C₃N₄QD has a typical thin 2D morphology, but one can not conclude so based on the TEM and AFM shown in Fig. S1~2. The lateral and the vertical dimensions of the particle are at a comparable level, which doesn't suggest a thin, 2D morphology. For example, see Fig. 6 for morphology of a more conventional 2D material at DOI: 0.1002/chem.200902478.

2. What percentage of the Zn ions are influenced by the CNQDs in terms of solvation structures? The concentration of CNQD is comparatively low, i.e., 0.1-4 mg/mL CNQD in 2M ZnSO₄; assuming 100% of the CNQDs are interacting with Zn cations, what percentage of Zn in the electrolyte will be coordinated by CNQD?

3. The authors claim that, "a 10 nm CNQD is able to carry up to 185 Zn²⁺." This agglomeration apparently makes the transport more sluggish and less uniform. This process can be thought of as incomplete dissociation. In this case, the transport of Zn ions (originally <1nm) needs to be coupled with >10 nm object. Do the authors expect the diffusion energy barrier of this coordinated object to be lower than normal, well dissociated Zn²⁺? Zn tends to develop porous structures if the transport is slower (PhysRevLett.56.1264; sciadv.abe0219). The authors should provide strong, direct support for this claim.

4. Fig. 3E – I suggest that the authors remove the dashed line in the original Zn-electrolyte interface since it blocks the view. It is hard to see the evolution of the surface accumulation of CNQD.

5. Fig. 4 – is the plating/stripping performed and then measured on Zn foil? If so, the textures identified may be dominated by the original texture of the Zn foil used. Why does the CNQD's peak disappear over cycling? The presentation of the GIXRD data needs to be improved – currently it's not very clear.

6. WRT the cathode, it is interesting that the cathode c-axis shrinks upon intercalation. Could the protons be intercalated, which is smaller than Zn²⁺, but also introduces some positive charges. Is it possible to intercalate large cations Zn²⁺ while having an even smaller c-axis lattice parameter? Does CNQD accumulate on the cathode (given that the authors assume a part of Zn²⁺ is coordinated with CNQD, meaning Zn²⁺ flux towards the cathode upon initial discharge should also deposit some CNQD?)?

7. Could this strategy work for other metal deposition systems such as Li, which also suffer from nonuniform deposition? A dynamic interphase could be helpful in solving these other problems(nano.2017.16; C9CS00883G).

Reviewer #2 (Remarks to the Author):

Comments:

In this work, C₃N₄QDs additive was introduced into aqueous electrolyte to inhibit side reaction and dendrites growth. MD simulation and DFT calculation demonstrate the changes of solvation structure and strong binding energy toward Zn. Meanwhile, the advanced characterization

techniques such as XANES spectra, in situ AFM, in situ fluorescence microscopy, GIXD and in situ Raman spectra were carried out deeply explore the action mechanism of C3N4QDs. However, electrochemical performance of both half cells and full cells are not excellent compared recent literature. Therefore, this work should be carefully evaluated after the follow suggestion are taken into consideration.

1. Carbon quantum dots are known as zero-dimensional carbon nanomaterials. You mentioned that C3N4QDs have a typical thin 2D morphology. Although AFM measurement indicates layer-structure, well-defined lattice fringes are not clearly observed in TEM image. The description may not suitable. If possible, please give related reference.
2. MD simulation was carried out to demonstrate the changes of solvation structure of Zn²⁺. How does C3N4QDs with an average lateral size of ~10 nm enter primary solvation shell of Zn²⁺? And what is the basis of molecular models of C3N4QDs? Related reference should be supplemented to confirm this conclusion.
3. As shown in Fig S8, when the concentration of C3N4QDs is 4 mg/mL, the smaller charge transfer resistance is obtained. Please give reasonable excuse.
4. Authors claimed that C3N4QDs can participate in metal coordination via intrinsic pore, therefore 185 Zn²⁺ ions are carried up at the same time. The pore structure of C3N4QDs should be given. Most important, this may contradict the previous MD calculation. The mechanism of C3N4QDs seems confusing, the detailed explanation should be added.
5. Real-time SEI reconstruction mechanism is demonstrated by in situ fluorescence microscopy observations. However, the fluorescent signal may not fully prove self-adaption and self-healing mechanism. The XPS results of electroplating and stripping to further demonstrate the conclusion.
6. SEM images indicate the successful suppression of Zn dendrites; however, cross-sectional images may more validly reflect morphology evolution of Zn.
7. The authors mentioned that C3N4QDs could effectively improve Zn²⁺ transfer number. And then you claimed that C3N4QDs slightly lowers the ionic conductivity and increase charge transfer resistance. Is there a necessary connection between the ionic conductivity and transference number?
8. The Zn||Zn symmetric cell with C3N4QDs exhibits excellent performance, however, the Zn||MnO₂ and Zn||VOPO₄ show poor cycling performance. And the rate performance of full cells is missing. Most important, the obvious activation process of Zn| C3N4QDs |VOPO₄ could be observed, but the same phenomena is not found in blank group.
9. The electrolyte of half cells and full cells are different, especially for Zn||VOPO₄.As we all know, electrolyte as vital component plays important role in electrochemical behavior. Different electrolyte means different property; therefore, it is necessary to use the same electrolyte.

REVIEWER COMMENTS

Reviewer #1 (Remarks to the Author):

Dear editor,

I read with interest the manuscript by Zhang et al. reporting on using a dynamic interphase formed by graphitic carbon nitride as an electrolyte additive to improve the performance of Zn metal anodes. The concept appears to be scientifically. The authors also used complementary characterization tools (those at beamlines) to investigate the process. Overall, this work appears to be of interest to the readership of Nature Comm, and I thus suggest that authors further revise the manuscript to address the issues as detailed below.

1. The authors claim the C₃N₄QD has a typical thin 2D morphology, but one can not

conclude so based on the TEM and AFM shown in Fig. S1~2. The lateral and the vertical dimensions of the particle are at a comparable level, which doesn't suggest a thin, 2D morphology. For example, see Fig. 6 for morphology of a more conventional 2D material at DOI: 0.1002/chem.200902478.

Response: We would like to thank the Reviewer for pointing out this somewhat confusing definition about 2D quantum dots. By definition, “dots” imply zero-dimension equiaxed structures like spheres or cubes, controversial to the expression of 2D. Strictly speaking, there is no 2D quantum dots. As the Reviewer pointed out, 2D morphologies are typically used to describe thin (few-layer) materials with large lateral sizes, such as a piece of few-layer graphene with a lateral to axial (thickness) size ratio of over 10,000.

The situation becomes interesting when the lateral size of a piece of 2D material (e.g., graphene) was reduced to several nanometers, which is in the range of the size of quantum dots and therefore often called quantum dots. As the lateral size/thickness ratio decreases to ~10, technically the resulting materials are not typical 2D morphology anymore. However, the terminology of “2D quantum dots” is widely used in literature to describe quantum dots derived from 2D materials and are not equiaxed, e.g., graphene QDs and MoS₂ QDs. Below list a few reviewer papers about this topic.

- Quantum dots derived from two-dimensional materials and their applications for catalysis and energy, *Chemical Society Reviews*, 2016, 45, 2239-2262
- Two-dimensional quantum dots: Fundamentals, photoluminescence mechanism, and their energy and environmental applications, *Materials Today Energy*, 2018, 10, 222-240
- A critical review on two-dimensional quantum dots (2D QDs): From synthesis toward applications in energy and optoelectronics, *Progress in Quantum Electronics*, 2019, 68, 100226

In the original manuscript, we followed the expression in these review papers and described our C₃N₄QDs as 2D QDs and with 2D morphology (they are not equiaxed

with lateral/thickness ratio of 5-10). After considering the Reviewer's comments, we agree that 2D QDs are strictly not an accurate terminology. Our materials are more like nanoplates, or nanotiles with quantum effects.

In the revised manuscript, we use "quantum nanotiles" to replace "2D QDs" and use "layered structure" to replace "2D morphology. We believe that these are more precise expressions. We would like to thank the Reviewer again for helping us clarify the confusing expression.

The relevant discussion has been added to the revised manuscript. (Page 4, line 5-9)

2. What percentage of the Zn ions are influenced by the CNQDs in terms of solvation structures? The concentration of CNQD is comparatively low, i.e., 0.1-4 mg/mL CNQD in 2M ZnSO₄; assuming 100% of the CNQDs are interacting with Zn cations, what percentage of Zn in the electrolyte will be coordinated by CNQD?

Response: We sincerely thank the reviewer to mention this critical point, which actually further strengthens the importance of the dynamic reconstructed process. The percentage of the Zn ions that are influenced by C₃N₄QDs was calculated based on a simplified model, where the diameter of the C₃N₄QDs is 10 nm and their thickness is 1.8 nm (based on the TEM and AFM analysis); each structural pore on C₃N₄QDs can adsorb one Zn²⁺ ion. As listed in Supplementary Table 2, even in this ideal situation, there will be only ~0.063% of Zn²⁺ ions coordinated with C₃N₄QDs in electrolyte with 0.5 mg mL⁻¹ C₃N₄QDs.

This extremely small percentage suggests that the impact of C₃N₄QDs on the overall Zn²⁺ solvation structure would not be significant without the formation of dynamic protective interphase. If we look at the distribution of the Zn²⁺ ions across the Zn symmetric cell at a certain time, the ones coordinated with C₃N₄QDs will be highly concentrated on the plating electrode surface where the protective interphase lies (Figure 3e), and the ones in the bulk electrolyte (not near the surface) will be mostly "free" (not coordinated with C₃N₄QDs). The C₃N₄QDs then constructed interphase that keeps the pores for ion-sieving open to enable water-free, single Zn²⁺ ion

conduction, that is, all of the deposited Zn atom (in the form of ions) must be coordinated with C₃N₄QDs protective interphase right before reaching the electrode and being reduced into Zn metal.

Hence the key role of the C₃N₄QDs additive is the construction of protective interphase upon the Zn anode.

The relevant discussion has been added to the revised manuscript. (Page 10, line 18-23)

Supplementary Table 2. The percentage of the Zn²⁺ions that interacted with C₃N₄QDs in 2M ZnSO₄ theoretically.

The concentration of C ₃ N ₄ QDs	The percentage of the Zn ²⁺ ions that interacted with C ₃ N ₄ QDs
0.1 mg mL ⁻¹	0.013%
0.5 mg mL ⁻¹	0.063%
1 mg mL ⁻¹	0.125%
2 mg mL ⁻¹	0.251%
3 mg mL ⁻¹	0.376%

3. The authors claim that, “a 10 nm CNQD is able to carry up to 185 Zn²⁺.” This agglomeration apparently makes the transport more sluggish and less uniform. This process can be thought of as incomplete dissociation. In this case, the transport of Zn ions (originally <1nm) needs to be coupled with >10 nm object. Do the authors expect the diffusion energy barrier of this coordinated object to be lower than normal, well dissociated Zn²⁺? Zn tends to develop porous structures if the transport is slower (PhysRevLett.56.1264; sciadv.abe0219). The authors should provide strong, direct support for this claim.

Response: That is a very interesting question, As discussed in the response to comment #2, only ~0.063% of Zn²⁺ ions are coordinated with C₃N₄QDs even in an ideal situation, and this fraction of Zn²⁺ are heavily concentrated in the C₃N₄QDs protective interphase on the Zn anode. In that case, the Zn²⁺ ions in the electrolyte are

almost entirely in the form of free Zn^{2+} ions solvated with water until the Zn^{2+} reaches the $\text{C}_3\text{N}_4\text{QDs}$ protective interphase. Hence the kinetics of the water-based electrolyte is barely affected as long as the $\text{C}_3\text{N}_4\text{QDs}$ protective interphase is established in each plating (not much movement of $\text{C}_3\text{N}_4\text{QDs}$).

However, at the beginning of each plating/stripping process (when the current just rotates directions), the $\text{C}_3\text{N}_4\text{QDs}$ protective interphase on one electrode will redisperse in the electrolyte, and migrate to the other electrode and form a protective interphase. This process involves the movement of $\text{C}_3\text{N}_4\text{QDs}$ that are coordinated with numbers of Zn^{2+} . As the Reviewer mentioned, compared to free Zn^{2+} ions solvated with water, the transport of the coordinated object would most likely be more sluggish. That is a clear disadvantage. The advantage is that a large amount of Zn^{2+} undergoes synchronous migration alongside the $\text{C}_3\text{N}_4\text{QDs}$ under the electrical field. All the Zn^{2+} ions on one $\text{C}_3\text{N}_4\text{QD}$ reach the electrode at the same time in a well-distributed mode (following the pattern of periodic coplanar zincophilic pores in $\text{C}_3\text{N}_4\text{QD}$). Such a transport behavior ensures a generous and homogeneous Zn^{2+} ion flux. The overall effect will be an interplay between these two factors. It needs to be emphasized that the formation of the protective interphase is complete in minutes judged by the luminescence microscope analysis. Most of the time in a cycle, $\text{C}_3\text{N}_4\text{QDs}$ are immobilized in the interface. Hence, the sluggish transport of coordinated objects in the first few minutes has a limited impact on the overall plating of Zn. That may explain the smooth and none porous deposition of Zn.

After the $\text{C}_3\text{N}_4\text{QDs}$ accumulate and construct the protective interphase on the surface of Zn anode, it serves as an ion-sieving film that prevents the reduction of solvated water and anions. The transference number ($t_{\text{Zn}^{2+}}$) was calculated to quantitatively describe the Zn^{2+} conducting ability of the $\text{C}_3\text{N}_4\text{QDs}$ protective interphase. A rather low $t_{\text{Zn}^{2+}}$ of 0.577 was obtained in the Zn symmetric cell under pure ZnSO_4 electrolyte owing to the faster migration speed of the anions than solvated Zn^{2+} , which is consistent with a previous report. $t_{\text{Zn}^{2+}}$ can be dramatically improved to 0.796 after the formation of $\text{C}_3\text{N}_4\text{QDs}$ interphase, (Supplementary Figure 9, Table 1). The structural periodic coplanar zincophilic pores in $\text{C}_3\text{N}_4\text{QDs}$ provide the active sites or solvating

groups for ion transfer, and the dense C₃N₄QDs interphase can block solvated water and anions from diffusing through this interphase. The high $t_{\text{Zn}^{2+}}$ eliminates the large Zn²⁺ concentration gradient and facilitates uniform ion distribution, resulting in homogeneous Zn plating.

The relevant discussion has been added to the revised manuscript. (Page 6, line 3-7; Page 6, line 8-12)

4. Fig. 3E – I suggest that the authors remove the dashed line in the original Zn-electrolyte interface since it blocks the view. It is hard to see the evolution of the surface accumulation of CNQD.

Response: Thanks a lot for your kind suggestions, the mark in Figure 3e has been changed to a small arrow in case the mark blocks the view.

Figure 3 has been revised in the revised manuscript.

5. Fig. 4 – is the plating/stripping performed and then measured on Zn foil? If so, the textures identified may be dominated by the original texture of the Zn foil used. Why does the CNQD's peak disappear over cycling? The presentation of the GIXRD data needs to be improved – currently it's not very clear.

Response: That is a good question. Indeed, the original texture of the Zn foil may dominate the results of many X-ray-based analyses, e.g., XRD. To avoid that, we utilized 2D synchrotron grazing-incidence X-ray diffraction (GIXD) to study the Zn foil electrodes after various plating/stripping processes. The advantage of GIXD technique on flat Zn electrodes is the limited penetration depth of the X-rays into the samples, with the benefit of low background scattering from the substrate. By varying the incident angle, the X-rays' penetration depth can be changed from a few nanometers up to 100 nanometers. As for our experiment, the GIXD patterns were measured in the incident angle range of 0.2° to 2.6°, and the ones measured at 0.8° are presented in Fig 4. The beam size is $\sim 5 \times 7 \mu\text{m}$ to collect the surface texture of Zn electrode that gets rid of the influence of bulk Zn foil. The scattered X-rays at 2θ

angular range of 6–73° are recorded by a 2D X-ray sensitive detector.

As can be seen from the SEM analysis (Figure 4i-4o), the surface of Zn electrode turns to be relatively rough and porous after the plating/stripping processes. Hence the X-rays' penetration depth enlarged and the intensity of Zn scattering signal increased sharply. The scattering signal of the C₃N₄QD is not as evident as that obtained after 1st plating, however, it still can be recognized if compared with that of Zn electrode obtained in pure ZnSO₄ electrolyte.

The relevant discussion has been added to the revised manuscript. (Page 12, line 24-28; Page 23, line 27-29; Page 24, line 1-5)

6. WRT the cathode, it is interesting that the cathode c-axis shrinks upon intercalation. Could the protons be intercalated, which is smaller than Zn²⁺, but also introduces some positive charges. Is it possible to intercalate large cations Zn²⁺ while having an even smaller c-axis lattice parameter? Does CNQD accumulate on the cathode (given that the authors assume a part of Zn²⁺ is coordinated with CNQD, meaning Zn²⁺ flux towards the cathode upon initial discharge should also deposit some CNQD?)?

Response: Many thanks. Actually, the reviewer points out one of the most critical challenges for the cathode materials in aqueous batteries, which is the protonation (ACS Sustainable Chem. Eng. 2021, 9, 8, 3223–3231). For the 2D VOPO₄ cathode, a large *d*-space of 4.2 Å makes both the protons and Zn²⁺ ions can intercalate into the lattice of VOPO₄ during the discharging process (Angew. Chem. Int. Ed. 2018, 57(37): 11978-11981). This leads to either serious voltage or capacity decay.

Additional XPS analysis was performed to evaluate the behavior of C₃N₄QDs on the cathode. The C₃N₄QDs additive was found also to accumulate on the VOPO₄ cathode constructing protective interphase, as illustrated in Supplementary Figure 40. The VOPO₄ cathode after the 15th cycle displays an evident N 1s signal indicative of the formation of the C₃N₄QDs interphase. This protective C₃N₄QDs interphase also induces an ion-sieving effect to enable single Zn²⁺ conduction, which can restrain the water activity and water-induced side reaction to a certain extent, significantly

reducing the dissolution of VOPO₄. Most importantly, the protonophilicity of C₃N₄ (J. Am. Chem. Soc., 2009, 131, 50-51.) favors capturing the protons thereby downgrading the protonation penetration depth. Combining the rather stiff nature of C₃N₄ (Adv. Mater., 2020, 32, 1908140), the formation of C₃N₄QDs interphase might maintain a rigid structure of VOPO₄ during insertion and deinsertion, hence preserving a stable cycling performance.

We would like to thank the reviewer for bringing up the potential impact on cathodes. From the XPS results and the information in the literature, we believe that the C₃N₄QDs should have a positive impact on cathodes and the impact could vary depending on the type of cathodes, e.g., MnO₂, VOPO₄, and V₂O₅. This communication manuscript is focusing on the dendrite elimination on the anode. The full cell experiments coupled with MnO₂, VOPO₄, and V₂O₅ cathodes are for the purpose of demonstrating our dendrite elimination strategy in a broad range of aqueous Zn-ion full cell configurations. In the planned follow up work, we will systematically investigate the impact of C₃N₄QDs on various cathodes and optimize the structure of C₃N₄QDs for each cathode

The relevant discussion has been added to the revised manuscript. (Page 19, line 13-21)

Supplementary Figure 40. (a)XPS survey spectra and (b)high-resolution N 1s spectra of the VOPO₄-cathode after the 15th cycle.

7. Could this strategy work for other metal deposition systems such as Li, which also

suffer from nonuniform deposition? A dynamic interphase could be helpful in solving these other problems(nnano.2017.16; C9CS00883G).

Response: Many thanks for the reviewer's valuable suggestion. Lithium metal is the ideal anode for the next generation of high-energy-density batteries. Continuous research on Li metal stripping/plating has deepened our understanding of the process but has not helped in solving the dendrite growth, side reactions, and infinite relative volume change in an effective manner (nnano.2017.16; C9CS00883G). The concept of our dynamic reconstructed solid-electrolyte interphase in this work might provide a brand-new route for accelerating the development of lithium metal anode.

Following the Reviewer's suggestion, we performed several preliminary tests in lithium symmetric cells. The C₃N₄QDs were then used as the electrolyte additives for lithium metal anode protection. The electrolyte contains 1 M LiTFSI in DOL/DME solution (v/v = 1:1) and 0.5 mg mL⁻¹ C₃N₄QDs. In Li–Li symmetric cell, the Li plates with a 15.8 mm diameter and a thickness of ~450 μm were used as working and counter electrodes. The electrochemical performance test was carried out in a CR2032-type coin cell by using lithium plate electrode, electrolyte, and polypropylene separators (Celgard 2500) in an Ar-filled glove box with water and oxygen levels below 0.5 ppm. As expected, the presence of C₃N₄QDs brought an enhanced plating/stripping cycling under 1 mA cm⁻²/1 mAh cm⁻². The pristine Li||Li cell delivers a severe polarization after 260th stripping/plating process attributed to the continuous electrolyte decomposition and dead Li formation, while Li|C₃N₄QDs|Li cell exhibits cyclic stability over 800 cycles. However, the C₃N₄QDs were found in situ anionic cross-linked and constructed a multifunctional SEI upon lithium anode, since a negative reduction potential of Li/Li⁺. To realize the novel dynamic reconstructed process, there are still a couple of issues that need to be addressed urgently.

Firstly, the dispersibility and stability of C₃N₄QDs in the organic solvent still need to be improved. Secondly, the addition of C₃N₄QDs might deteriorate the kinetic of lithium-ion diffusion. Last but not the least, the reduction of Li/Li⁺ occurs at -3.04 V vs. standard hydrogen electrode (SHE), where the electron transfers from metal Li to

the C₃N₄QDs might reduce the oxygen-containing groups at the edge of C₃N₄QDs, and induce a mild cross-linking reaction, resulting in the formation of permanent thin film on lithium metal's surface. This process might have a substantial impact on suppressing lithium dendrite growth, but this suppression is built on the gradual consumption of C₃N₄QDs additives that occurs during lithium deposition and is thereby not sustainable and falls short in long-term cycling. Of course, the dynamic reconstructed solid-electrolyte interphase mechanism is missing. Therefore, we decided to not include the Li metal protection results in the revised manuscript.

Nevertheless, we have devoted continuous endeavors to the dynamic reconstructed solid-electrolyte interphase upon lithium metal anode. On one hand, we are trying to modify the functional groups on C₃N₄QDs to strengthen their tolerance under such negative potential of Li/Li⁺ conversion. On the other hand, we are looking for other quantum dots additives (ZnIn₂S₄ and so on) that can circumvent all the above problems, hence investigating their possibility on building the dynamic reconstructed solid-electrolyte interphase.

Reviewer #2 (Remarks to the Author):

Comments:

In this work, C₃N₄QDs additive was introduced into aqueous electrolyte to inhibit side reaction and dendrites growth. MD simulation and DFT calculation demonstrate the changes of solvation structure and strong binding energy toward Zn. Meanwhile, the advanced characterization techniques such as XANES spectra, in situ AFM, in situ fluorescence microscopy, GIXD and in situ Raman spectra were carried out deeply explore the action mechanism of C₃N₄QDs. However, electrochemical performance of both half cells and full cells are not excellent compared recent literature. Therefore, this work should be carefully evaluated after the follow suggestion are taken into consideration.

1. Carbon quantum dots are known as zero-dimensional carbon nanomaterials. You mentioned that C₃N₄QDs have a typical thin 2D morphology. Although AFM measurement indicates layer-structure, well-defined lattice fringes are not clearly observed in TEM image. The description may not suitable. If possible, please give related reference.

Response: We would like to thank the Reviewer for pointing out this somewhat confusing definition about 2D quantum dots. By definition, “dots” imply zero-dimension equiaxed structures like spheres or cubes, controversial to the expression of 2D. Strictly speaking, there is no 2D quantum dots. As the Reviewer pointed out, 2D morphologies are typically used to describe thin (few-layer) materials with large lateral sizes, such as a piece of few-layer graphene with a lateral to axial (thickness) size ratio of over 10,000.

The situation becomes interesting when the lateral size of a piece of 2D material (e.g., graphene) was reduced to several nanometers, which is in the range of the size of quantum dots and therefore often called quantum dots. As the lateral size/thickness ratio decreases to ~10, technically the resulting materials are not typical 2D morphology anymore. However, the terminology of “2D quantum dots” is widely used in literature to describe quantum dots derived from 2D materials and are not equiaxed, e.g., graphene QDs and MoS₂ QDs. Below list a few reviewer papers about this topic.

- Quantum dots derived from two-dimensional materials and their applications for catalysis and energy, *Chemical Society Reviews*, 2016, 45, 2239-2262
- Two-dimensional quantum dots: Fundamentals, photoluminescence mechanism, and their energy and environmental applications, *Materials Today Energy*, 2018, 10, 222-240
- A critical review on two-dimensional quantum dots (2D QDs): From synthesis toward applications in energy and optoelectronics, *Progress in Quantum Electronics*, 2019, 68, 100226

In the original manuscript, we followed the expression in these review papers and

described our C_3N_4 QDs as 2D QDs and with 2D morphology (they are not equiaxed with lateral/thickness ratio of 5-10). After considering the Reviewer's comments, we agree that 2D QDs are strictly not an accurate terminology. Our materials are more like nanoplates, or nanotiles with quantum effects. Following the Reviewer's suggestion, we tried to observe the lattice fringes of the as-synthesized C_3N_4 QDs via HR-TEM. Compared to large C_3N_4 nanosheets, we noticed that it is extremely challenging to obtain good contrast since the C_3N_4 QDs have to be cast on TEM grid with a 3 nm amorphous carbon supporting film which dramatically impairs the contrast. While the large C_3N_4 nanosheets can be cast on a TEM grid with holey supporting film to avoid the impact of the carbon substrate. Despite the relatively poor contrast, we observed the two-dimensional lattices of C_3N_4 QDs (Figure Ra). Fast-Fourier transform (FFT) analysis of the image yielded a hexagonal pattern in the reciprocal space, indicating that the observing plane (perpendicular to the beam direction) is the basal (002) plane of C_3N_4 QDs (Figure Rb) (Microscopy and Microanalysis, 2016, 22: 1668-1669)

In the revised manuscript, we use “quantum nanotiles” to replace “2D QDs” and use “layered structure” to replace “2D morphology. We believe that these are more precise expressions. We would like to thank the Reviewer again for helping us clarify the confusing expression.

The relevant discussion has been added to the revised manuscript. (Page 4, line 5-9)

Figure R (a) HR-TEM image of the as-synthesized C_3N_4 QD, (b) FFT of the area in the red square. The scale bar in (a) is 5 nm.

2. MD simulation was carried out to demonstrate the changes of solvation structure of Zn^{2+} . How does C_3N_4QDs with an average lateral size of ~ 10 nm enter primary solvation shell of Zn^{2+} ? And what is the basis of molecular models of C_3N_4QDs ? Related reference should be supplemented to confirm this conclusion.

Response: Many thanks. The basic model of the C_3N_4QDs is established according to Lu's work (10.1039/c4tc02111h), and the structure of C_3N_4QDs is presented in Supplementary Figure 8a in Supporting information. As the reviewer mentioned, a Zn^{2+} solvated ion is much smaller than a C_3N_4QD , but profiting from the interaction between Zn^{2+} and subnanometric pores in C_3N_4QDs , the C_3N_4QDs could serve as the ion carrier that adsorbs Zn^{2+} ions. This process is thermodynamically favorable, as evident by the movement track of the Zn^{2+} ion in the $C_3N_4QDs-ZnSO_4$ electrolyte analyzed via MD simulation (Supplementary Figure 6). The intrinsic subnanometric pore in C_3N_4QD serves as the most stable binding site for the Zn^{2+} . The radial distribution functions (RDFs) in Figure 1e confirm the coexistence of $Zn^{2+}-O$ and $Zn^{2+}-N$ coordination in the $ZnSO_4-C_3N_4QDs$ system, with bond lengths of 2.05 Å and 3.35 Å, respectively, demonstrating again the changes of solvation structure of Zn^{2+} via $Zn^{2+}-N$ interaction. Additionally, the obvious blueshift of the vibration stretching of SO_4^{2-} in $C_3N_4QDs-ZnSO_4$ electrolyte on FT-IR spectra in Supplementary Figure 5 unveils lower binding of SO_4^{2-} and thereby confirms further separation from the Zn^{2+} coordination sheath. The redshift of the Zn-O signal on EXAFS in Figure 1c verified the weakened interaction between Zn^{2+} and H_2O and a reduced O-coordination around Zn^{2+} .

The changes of Zn^{2+} solvation structure may be considered as large C_3N_4QDs entering a much smaller primary solvation shell of Zn^{2+} . But a more precise description would be that the intrinsic subnanometric pore in C_3N_4QD (more specifically the N atoms around a pore) enter the primary solvation shell of Zn^{2+} .

The relevant discussion has been added to the revised manuscript. (Page 5, line 8-12; Page 24, line 13-14)

Supplementary Figure 6. (a) The position motion track and (b) projection motion track of Zn^{2+} ions relative to C_3N_4QD after 30000fs in MD simulation.

3. As shown in Fig S8, when the concentration of C_3N_4QDs is 4 mg/mL, the smaller charge transfer resistance is obtained. Please give reasonable excuse.

Response: We sincerely thank the reviewer mention it and are extremely sorry for our carelessness, the arrangement of the figures in Supplementary Figure 8 was not updated properly after repeated organization. We have corrected it carefully and now all the figures in Supplementary Figure 9 are in the correct order. Each of the figures was also assigned the correct label.

The misarrangement occurs because we first organized them by the order of 2 M $ZnSO_4$ + 0.1 mg ml^{-1} C_3N_4QDs , 2 M $ZnSO_4$ + 0.5 mg ml^{-1} C_3N_4QDs , 2 M $ZnSO_4$ + 1 mg ml^{-1} C_3N_4QDs , 2 M $ZnSO_4$ + 2 mg ml^{-1} C_3N_4QDs , 2 M $ZnSO_4$ + 4 mg ml^{-1} C_3N_4QDs and the pristine 2 M $ZnSO_4$, after reconsidering, we arranged the i-t curves of pristine 2 M $ZnSO_4$ from the end to the first, but forgot to arrange the second column of EIS plots. We have provided the original misarranged Supplementary Figure 8 alongside the corrected one, only the arrangement was changed. All the calculations were calculated based on the corrected values.

With increasing the concentration of C_3N_4QDs , the charge transfer resistance is slightly enlarged. This might result from the slightly decreased ionic conductivity after the addition of C_3N_4QDs (Supplementary Figure 23). However, the charge transfer resistance upon standing in $ZnSO_4$ electrolyte progressively increases, which

reflects the passivation of Zn anode, whereas it can stay almost constant in ZnSO₄-C₃N₄QDs (Supplementary Figure 27, Figure 28). The high interfacial stability also in mechanical terms provides homogeneous Zn²⁺ conduction pathways and accommodates volume change during the harsh electrochemical process.

Based on the initial and steady-state resistance, we calculated the transference number of Zn²⁺ ($t_{\text{Zn}^{2+}}$) in various electrolytes, as listed in Supplementary Table 1. A rather low $t_{\text{Zn}^{2+}}$ of 0.577 was obtained in the Zn symmetric cell under pure ZnSO₄ electrolyte owing to the faster migration speed of the anions than solvated Zn²⁺, which is consistent with a previous report. $t_{\text{Zn}^{2+}}$ can be dramatically improved to 0.796 in ZnSO₄-C₃N₄QDs, where the protective C₃N₄QDs interphase can be constructed during the plating process. The structural periodic coplanar zincophilic pores in C₃N₄QDs provide the active sites or solvating groups for ion transfer, and the dense C₃N₄QDs interphase can block solvated water and anions from diffusing through this interphase. The high $t_{\text{Zn}^{2+}}$ contributes to eliminating the large Zn²⁺ concentration gradient and facilitating uniform ion distribution, resulting in homogeneous Zn plating.

Original Supplementary Figure 8. (a) Current-time plots of Zn symmetric cell in various electrolytes after polarization at a constant potential (25 mV) for 2000 s, (b) the impedance spectra before and after the polarization.

Corrected Supplementary Figure 9. (a) Current-time plots of Zn symmetric cell in various electrolytes after polarization at a constant potential (25 mV) for 2000 s, (b) the impedance spectra before and after the polarization.

Supplementary Table 1. Calculated $t_{Zn^{2+}}$ in different ZnSO₄-based electrolytes.

Electrolytes	$t_{Zn^{2+}}$
2M ZnSO ₄	0.577
2M ZnSO ₄ + 0.1 mg ml ⁻¹ C ₃ N ₄ QDs	0.610
2M ZnSO ₄ + 0.5 mg ml ⁻¹ C ₃ N ₄ QDs	0.712
2M ZnSO ₄ + 1 mg ml ⁻¹ C ₃ N ₄ QDs	0.796
2M ZnSO ₄ + 2 mg ml ⁻¹ C ₃ N ₄ QDs	0.674
2M ZnSO ₄ + 4 mg ml ⁻¹ C ₃ N ₄ QDs	0.682

4. Authors claimed that C₃N₄QDs can participate in metal coordination via intrinsic pore, therefore 185 Zn²⁺ ions are carried up at the same time. The pore structure of C₃N₄QDs should be given. Most important, this may contradict the previous MD calculation. The mechanism of C₃N₄QDs seems confusing, the detailed explanation should be added.

Response: Many thanks. A typical C₃N₄QD structure was presented in Supplementary Figure 1, which is composed of the condensed tri-s-triazine (tri-ring of C₆N₇) subunits connected through planar tertiary amino groups, hence possessing periodic pores of ~0.68 nm in the lattice. As characterized by TEM (Supplementary Figure 1) and AFM (Supplementary Figure 2), the as-synthesized C₃N₄QD in this work shows an average lateral size of ~10 nm, thereby constituting ~90 subnanometric pores. These subnanometric pores are identified as the Zn²⁺ ions' adsorption sites.

In the MD simulation of Figure 1d, we choose a typical periodic unit of C₃N₄QD, one subnanometric pore surrounded by three tri-s-triazine, instead of a 10 nm C₃N₄QD, to constitute the cubic box of ZnSO₄-C₃N₄QD electrolyte. The strong interaction between C₃N₄QD and Zn²⁺ ions renders the C₃N₄QD to enter the primary solvation shell of Zn²⁺ and replace two water molecules forming a [Zn(C₃N₄)(H₂O)₄]²⁺ complex. The intrinsic subnanometric pore in C₃N₄QD serves as the most stable binding site for the Zn²⁺. The radial distribution functions (RDFs) (Figure 1e) confirm the coexistence of Zn²⁺-O and Zn²⁺-N coordination in ZnSO₄-C₃N₄QD system, with bond lengths of

2.05 Å and 3.35 Å, respectively. The Zn²⁺-O pair in ZnSO₄-C₃N₄QDs is slightly enlarged compared with that in ZnSO₄ (~1.94 Å), in accordance with the XANES results in Figure 1b. The weakened bonding strength between Zn²⁺ and H₂O reduces the proton activity and suppresses electrochemical water decomposition, as also evidenced by a gradually lower hydrogen evolution potential with increasing doses of C₃N₄QD in the electrolyte (Supplementary Figure 7).

The relevant discussion has been added to the revised manuscript. (Page 4, line 5-9; Page 5, line 8-12)

Supplementary Figure 1. The typical pore structure of C₃N₄QD, the grey, and blue balls represent C and N atoms.

5. Real-time SEI reconstruction mechanism is demonstrated by in situ fluorescence microscopy observations. However, the fluorescent signal may not fully prove self-adaption and self-healing mechanism. The XPS results of electroplating and stripping to further demonstrate the conclusion.

Response: Thank you for your suggestions. The surface chemistry of Zn electrode is evaluated by X-ray photoelectron spectroscopy (XPS) analysis. The Zn anodes after 15th stripping/plating process in symmetric cells using ZnSO₄ and ZnSO₄-C₃N₄QDs electrolyte (labeled as Zn@ZnSO₄ and Zn@ZnSO₄-C₃N₄QDs, respectively) are collected and characterized. The Zn@ZnSO₄ displays a distinct sulfate signal (Supplementary Figure 13), indicative of the formation of a Zn₄SO₄(OH)₆·xH₂O

deposit insulated the Zn^{2+} ion flux. This finding illustrates the passivation of Zn electrode occurring in conventional ZnSO_4 electrolytes. In the case of $\text{Zn}@Zn\text{SO}_4\text{-C}_3\text{N}_4\text{QDs}$, no sulfate component can be detected, providing strong evidence for the suppression of corrosion and the ion-sieving effect of the $\text{C}_3\text{N}_4\text{QDs}$ interphase. Moreover, the high-resolution C1s and N1s XPS spectra on $\text{Zn}@Zn\text{SO}_4\text{-C}_3\text{N}_4\text{QDs}$ show evident C=C (284.6 eV), C=N (285.6 eV), C-N (287.9 eV), C-N=C (398.6 eV), and N-(C)₃ (399.7 eV) components, which are all well in line with the XPS peaks of the original $\text{C}_3\text{N}_4\text{QDs}$, demonstrating that $\text{C}_3\text{N}_4\text{QDs}$ interphase with good structural integrity is formed (Figure 3a, 3b). Further Ar^+ sputtering XPS depth profiling in Figures 3c and 3d unveil that the chemical composition of C 1s and N 1s components remain almost unchanged for 600 s sputtering (0.1 nm s^{-1}), suggesting a tiled multilayer structure of the SEI and confirming the $\text{C}_3\text{N}_4\text{QDs}$ are efficiently involved in the Zn electro-deposition process. The *in situ* constructed $\text{C}_3\text{N}_4\text{QDs}$ interphase on Zn electrode is up to 100 nm thick and considered internally stacked sheets bound *via* van der Waals attractions. It should be noted that the nitrogen content in the stripping side of $\text{Zn}@Zn\text{SO}_4\text{-C}_3\text{N}_4\text{QDs}$ is found strikingly less than that in plating, attributed to the dynamic redispersion process of the $\text{C}_3\text{N}_4\text{QDs}$ interphase under the positive surface potential of the applied electrical field. This promising SEI reconstruction process implies self-adaption and self-healing, effectively maintaining constant conformal contact with Zn anode, timely correcting the Zn plating behavior, and intrinsically eradicating the irreversible fracture of the protective interphases, thereby guaranteeing a stable operation of Zn anode

The relevant discussion has been added to the revised manuscript. (Page 9, line 9-23; Page 10, line1-6)

6. SEM images indicate the successful suppression of Zn dendrites; however, cross-sectional images may more validly reflect morphology evolution of Zn.

Response: Many thanks. Focused ion beam (FIB)-SEM analysis was conducted to provide insights into the inner microstructural details of Zn growth in the different

electrolytes. As displayed in Supplementary Figure 20, the cycled Zn@ZnSO₄ and Zn@ZnSO₄-C₃N₄QD electrodes were first milled in a cavity by FIB. Then, the electrodeposited Zn morphology could be observed in the side-view SEM images. Hexagonal Zn platelets with various sizes are observed to form loosely connected building blocks assembled to create a porous electrodeposit structure on Zn@ZnSO₄. The Zn deposits are revealed to be randomly oriented. In contrast, the Zn@ZnSO₄-C₃N₄QD electrode exhibits dense Zn electrodeposits without inner interstices, no preferred oriented growth is detected and uniform nanoscale surface fluctuation is formed. This phenomenon is in line with the in-situ AFM observations, which might be resulting from the construction of the C₃N₄QD protective interphase upon Zn anode, inducing the ion-sieving effect and modulating the Zn electrodepositing behavior.

The relevant discussion has been added to the revised manuscript. (Page 13, line 12-19)

Supplementary Figure 20. Focused ion beam (FIB)-SEM images of (a) Zn@ZnSO₄

after 15 cycles, (b) Zn@ZnSO₄-C₃N₄ after 15 cycles, (c) Zn@ZnSO₄ after 50 cycles, (d) Zn@ZnSO₄-C₃N₄ after 50 cycles, respectively.

7. *The authors mentioned that C₃N₄QDs could effectively improve Zn²⁺ transfer number. And then you claimed that C₃N₄QDs slightly lowers the ionic conductivity and increase charge transfer resistance. Is there a necessary connection between the ionic conductivity and transference number?*

Response: Thanks. In this manuscript, the ionic conductivity was employed to evaluate the properties of aqueous electrolyte with additive, and the Zn²⁺ transference number was to describe the Zn²⁺ conducting ability across the constructed C₃N₄QD interphase upon Zn anode.

Similar to Li-ion batteries, the notorious side reactions of Zn with water are self-enhancing in aqueous Zn-ion batteries. One of the key roles of the electrolyte additives is to build dense artificial interphase on the Zn metal surface to achieve side-reaction-free and dendrite-free Zn electrodeposition. Theoretically, the ideal protective interphase should meet the following criteria. First, it should be water-insoluble, which helps to block the reactive water molecules from encountering the Zn electrode surface. Second, it should feature a high ionic conductivity and a low electronic conductivity, promoting uniform Zn²⁺ deposition underneath the protective interphase. Third, it should have good adhesion to the Zn metal surface, so that this protective interphase would not be detached from the Zn surface during plating.

One of the key indexes for evaluating the aqueous electrolyte, the ionic conductivity, is measured after introducing C₃N₄QD in ZnSO₄ electrolyte. The ionic conductivity of ZnSO₄-C₃N₄QD is found to be slightly decreased since an insulated colloid component is involved. Hence the charge transfer resistances of metallic Zn under vast ZnSO₄-C₃N₄QD are enlarged. However, as for the electrolyte within 2 M ZnSO₄ + 0.5 mg ml⁻¹ C₃N₄QDs, the ionic conductivity only decreases 2.38% when compared with the pristine 2 M ZnSO₄. It can be reasonably speculated that the reaction kinetics of the water-based electrolyte is not weakened after the addition of C₃N₄QDs. Meanwhile, good ionic conductivity could facilitate Zn²⁺ diffusion through the

constructed C₃N₄QDs protective interphase.

On the other hand, the divalent Zn²⁺ ion (radius, 0.74 Å) has a much higher electric charge density compared to the monovalent Li⁺ and Na⁺ ions (radii, 0.76 and 1.02 Å, respectively), the energy barrier of Zn²⁺ transfer in the SEI layers should be considered. Therefore, the transference number ($t_{\text{Zn}^{2+}}$) in various ZnSO₄-C₃N₄QD system were calculated to quantitatively describe the Zn²⁺ conducting ability of the constructed C₃N₄QDs interphase. A rather low $t_{\text{Zn}^{2+}}$ of 0.577 was obtained in the Zn symmetric cell under pure ZnSO₄ electrolyte owing to the faster migration speed of the anions than solvated Zn²⁺, which is consistent with a previous report. $t_{\text{Zn}^{2+}}$ can be dramatically improved to 0.796 after the formation of C₃N₄QDs interphase, (Supplementary Figure 9, Table 1). The structural periodic coplanar zincophilic pores in C₃N₄QDs provide the active sites or solvating groups for ion transfer, and the dense C₃N₄QDs interphase can block solvated water and anions from diffusing through this interphase. The high $t_{\text{Zn}^{2+}}$ eliminates the large Zn²⁺ concentration gradient and facilitates uniform ion distribution, resulting in homogeneous Zn plating.

8. The Zn||Zn symmetric cell with C₃N₄QDs exhibits excellent performance, however, the Zn||MnO₂ and Zn||VOPO₄ show poor cycling performance. And the rate performance of full cells is missing. Most important, the obvious activation process of Zn/C₃N₄QDs/VOPO₄ could be observed, but the same phenomena is not found in blank group.

Response: Thanks for your suggestions. In this work, we employed the C₃N₄QDs electrolyte additive to induce a novel real-time reconstructed SEI mechanism mainly for figuring out the irreversibility and dendrite-growth issues of Zn anode. The Zn||SS asymmetric cell and the Zn||Zn symmetric cell were assembled to characterize the Coulombic efficiency and long-term galvanostatic Zn stripping/plating process under the introduction of C₃N₄QDs. As expected, the C₃N₄QDs additives not only modulate the solvent structure of Zn²⁺ ions but also assist the construction of a dynamic & self-repairing protective interphase, the Coulombic efficiency and Zn stripping/plating cycling have been greatly enhanced.

To demonstrate the applicability of the C₃N₄QDs additive alongside their real-time reconstructed SEI mechanism in aqueous Zn-ion batteries, the Zn-ion full cells were fabricated using a MnO₂, V₂O₅, and VOPO₄ cathode. Clear improvements are observed in the full cells with all three cathodes. However, the performance of Zn-ion full cells is much more complicated since it is related to both the anode and the cathode. Serious decomposition/dissolution of the conventional MnO₂, V₂O₅, and VOPO₄ cathode downgrade the cycling stability of the Zn-ion full cells. Comparing the Zn||Zn symmetric cell and full cells stabilities, it can be concluded that the performance degradation of full cells with C₃N₄QDs is dominated by the degradation of the cathodes. To further increase the performance of full cells, some innovations on the cathodes are essential. In the next step, we will systematically explore the strategies for alleviating the dissolution issues of Mn-based and V-based cathodes in aqueous Zn-ion batteries, alongside understanding their electrochemical behaviors. We are expecting a specifically tuned strategy for each cathode material.

The rate performance of Zn|C₃N₄QDs|MnO₂ and Zn|C₃N₄QDs|VOPO₄ full cells with reference to the corresponding Zn||MnO₂ and Zn||VOPO₄ full cells were illustrated in Supplementary Figure 36. The capacity of both Zn|C₃N₄QDs|MnO₂ and Zn||MnO₂ drastically attenuated with the increase of cycle number and current density, which is caused by the serious decomposition/dissolution of MnO₂ cathode (Energy Environ. Sci., 2022, 15, 1106–1118). The Zn|C₃N₄QDs|MnO₂ exhibits slightly enhanced capacity retention. As for the VOPO₄-based full cells, a high capacity of ~100 mAh g⁻¹ can be achieved at the current density of 0.1 A g⁻¹, attributing to the major capacity contribution stemming from the diffusion-controlled process. The Zn|C₃N₄QDs|VOPO₄ present good capacity retention with the increase of current as well as over cycling. After harsh 75 cycles, it can be noted that the Zn|C₃N₄QDs|VOPO₄ shows a recovered capacity of 115 mAh g⁻¹ when the current density is set back to 0.1 A g⁻¹. An obvious activation process and a stable profile were obtained. In contrast, the Zn||VOPO₄ gradually downgraded, which is in line with the cycling results in Figure 6d. This might imply that the C₃N₄QDs also play a positive role in advancing the deficiencies of decomposition/dissolution of VOPO₄

cathode in aqueous electrolytes. The C_3N_4 QDs could accumulate on $VOPO_4$ cathode to form protective cathode-electrolyte interphase, similar to the organic carbonaceous interphases upon an aqueous lithium cathode (Adv. Mater. 2020, 32, 2004017), facilitating stable interface chemistry and substantially inhibiting water decomposition on the cathode surface, hence restraining the decomposition/dissolution of $VOPO_4$ cathode. Whereas for pristine $VOPO_4$ cathode, the serious decomposition/dissolution of $VOPO_4$ renders the $Zn||VOPO_4$ full cell drastically attenuated. In the following study, we will provide an in-depth, mechanistic understanding of the roles of C_3N_4 QDs on $VOPO_4$ cathode, aiming to expedite the application of next-generation aqueous Zn-ion batteries.

The relevant discussion has been added to the revised manuscript. (Page 18, line 20-23; Page 19, line 5-9)

Supplementary Figure 36. (a) Rate performances of $Zn|C_3N_4QDs|MnO_2$ and $Zn||MnO_2$ from 0.1 to 5 C, (b) Rate performances of $Zn|C_3N_4QDs|VOPO_4$ and $Zn||VOPO_4$ from 0.1 to 5 A g⁻¹.

9. The electrolyte of half cells and full cells are different, especially for $Zn||VOPO_4$. As we all know, electrolyte as vital component plays important role in electrochemical behavior. Different electrolyte means different property; therefore, it is necessary to use the same electrolyte.

Response: Many thanks. This real-time reconstructed SEI process with the addition of C_3N_4 QDs was studied in $ZnSO_4$ electrolytes. After evaluating the merits of

C₃N₄QDs additive in Zn||Zn symmetric cell, we explored their application in Zn-ion full batteries composed of a metallic Zn anode and a MnO₂ cathode in the same ZnSO₄ electrolyte. Additional Mn²⁺ ions are introduced to the electrolyte to mitigate the dissolution of MnO₂ which is a common practice in MnO₂-based full cells.

To further demonstrate the role of C₃N₄QDs in ZnSO₄ electrolyte-based full cells, we also employed a V₂O₅ cathode with ZnSO₄ electrolyte to fabricate a Zn||V₂O₅ full cells in the revised manuscript. As displayed in Supplementary Figure 32, both Zn||V₂O₅ and Zn|C₃N₄QDs|V₂O₅ cells exhibit a gradual increase in capacity in the initial tens of cycles, which is accordant with many previous reports and is related to the activation of the Zn²⁺ intercalation process. The initial discharge capacity of Zn|C₃N₄QDs|V₂O₅ cell is 153 mAh g⁻¹, which increases to the maximum of 170 mAh g⁻¹ at the 80th cycle and remains 112 mAh g⁻¹ after 500 cycles, 85 mAh g⁻¹ after 1000 cycles. In contrast, the Zn||V₂O₅ cell can only retain a capacity of 80 mAh g⁻¹ after 500 cycles, 52 mAh g⁻¹ after 1000 cycles.

It is well known that the VOPO₄ cathode is less stable in the aqueous ZnSO₄ electrolyte, and the common ZnSO₄ electrolyte displays an O₂ evolution potential of about 2.0 V versus Zn/Zn²⁺ (Angew. Chem. Int. Ed., 2019, 58(46): 16358-16367). Therefore, we prepared 3 M aqueous Zn(OTf)₂ electrolyte with 0.5 mg ml⁻¹ C₃N₄QDs for a VOPO₄-based full cell test. The Zn|C₃N₄QDs|VOPO₄ full cell witnesses miraculously stable operation over 3000 cycles at 1 A g⁻¹ with a capacity retention of 86.1% (Figure 6d). But as the reviewer mentioned, the electrolyte is a vital component that plays important role in electrochemical behavior, and different electrolyte means different property. However, we feel that the VOPO₄ cathode system could demonstrate the diversity of the C₃N₄QDs and the universal reconstructed SEI mechanism in an aqueous electrolyte. If the reviewer feels that it is improper to include a full cell with Zn(OTf)₂ electrolyte, we could delete the part of the VOPO₄ cathode system, to make this study more focused.

The relevant discussion has been added to the revised manuscript. (Page 18, line 1-6)

Supplementary Figure 32. (a) Typical voltage profiles of 1st and 500th cycles of Zn||V₂O₅ and Zn||C₃N₄QDs|V₂O₅ batteries at 1 A g⁻¹; (b) Cyclic performance of Zn||V₂O₅ and Zn||C₃N₄QDs|V₂O₅ batteries at 1 A g⁻¹.

Reviewer comments, second round review

Reviewer #1 (Remarks to the Author):

The authors have taken solid steps to address the questions I raised in the first round of review. I now believe that the manuscript is suitable for publication in Nat. Comm. and that it will be a impactful contribution to the field.

Reviewer #2 (Remarks to the Author):

The revised version could be accepted for publication.